# LOGO — LONG cONTEXT aliGNMENT VIA EFFI-CIENT PREFERENCE OPTIMIZATION

## ABSTRACT

Long-context models (LCMs) have shown great potential in processing long in-put sequences (even more than 100M tokens) conveniently and effectively. With significant progress, recent research has pointed out that LCMs can accurately locate token-level salient information within the context. Yet, the generation per-formance of these LCMs is far from satisfactory and might result in misaligned responses, such as hallucinations. To enhance the generation capability of LCMs, existing works have investigated the effects of data size and quality for both pre-training and instruction tuning. Though achieving meaningful improvement, pre-vious methods fall short in either effectiveness or efficiency. In this paper, we in-troduce LOGO (Long cOntext aliGnment via efficient preference Optimization), a training strategy that first introduces preference optimization for long-context alignment. To overcome the GPU memory-bound issue caused by the long se-quence, LOGO employs a reference-free preference optimization strategy and adopts a position synthesis method to construct the training data. By training with only 0.3B data on a single 8×A800 GPU machine for 16 hours, LOGO allows the Llama-3-8B-Instruct-80K model to achieve comparable performance with GPT-4 in real-world long-context tasks while preserving the model's original capabilities on other tasks, e.g., language modeling and MMLU. Moreover, LOGO can extend the model's context window size while enhancing its generation performance.

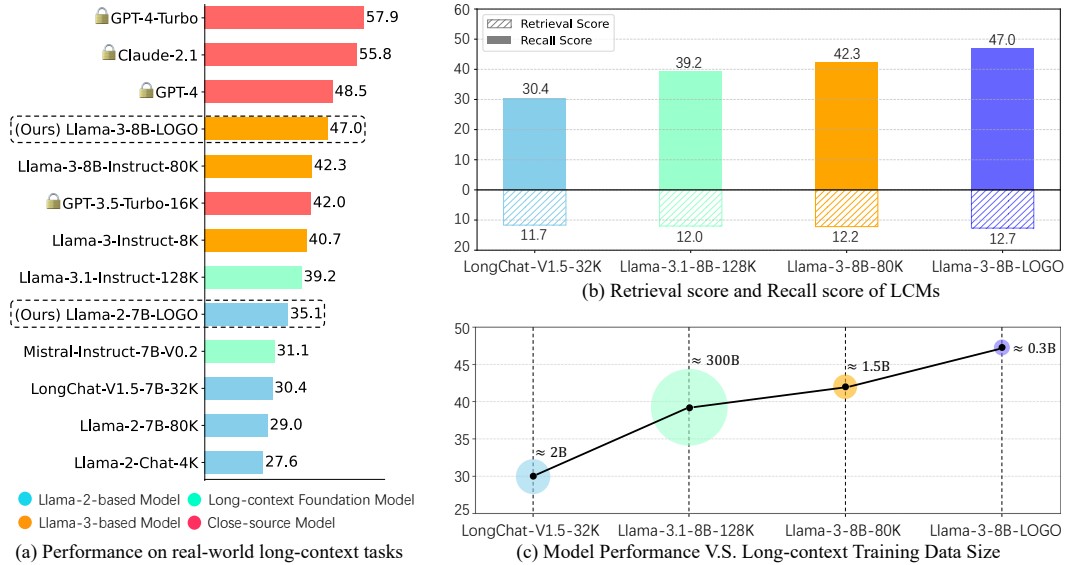

Figure 1: (a) Performance of LCMs on real-world long-context tasks; (b) Retrieval score (long-context understanding ability) and recall score (generation ability) of LCMs on the synthetic retrieval long-context task (multi-value NIAH); (c) Long-context (pre-)training data size for each LCM.

## 1 INTRODUCTION

With the rapid advancements of Large Language Models (LLMs), handling long contexts (even more than 100M tokens (anthropic, 2024)) has become a fundamental capability for recent LLMs. This

further unlocks the potential of LLMs for novel tasks and applications, e.g., code analysis (Zhu et al., 2024), while simultaneously eliminating the need for complex toolchains and intricate workflows that were previously required to overcome the context-length constraints (Ravaut et al., 2024).

Yet, recent studies have pointed out that these long-context models (LCMs) failed to achieve satisfactory performance in long-context tasks, where LCMs might produce misaligned results, such as instruction unfollowing and hallucinations (Belyi et al., 2024; Zhang et al., 2024a). To mitigate the above issue, the open-source community has made significant efforts, primarily focusing on building high-quality long instruction data and extending the data size (Wu et al., 2024a; Bai et al., 2024; Fu et al., 2024; Bai et al., 2024). As shown in Fig. 1, though achieving meaningful improvement, these methods fall short in effectiveness or efficiency. For instance, the Llama-3.1-8B-128K model AI@Meta (2024a) was pre-trained on around 300B long instruction data, but it even underperforms the Llama-3-8B-Instruct-80K model (Zhang et al., 2024b), which was post-trained with 1.5B high-quality long instruction data based on the Llama-3-8B-Instruct model (AI@Meta, 2024b). As for the Llama-3-8B-Instruct-80K model, it shows slight improvement compared to the baseline and still lags greatly behind the closed-source counterparts like GPT-4 (Achiam et al., 2023).

Recently, Wu et al. (2024b) pointed out that LCMs can accurately locate token-level salient information within the context. As shown in Fig. 1(b), we visualize the information retrieval capability[1] (reflected by the retrieval score) and the generation capability (reflected by the recall score) of different LCMs on the synthetic retrieval task, where we can observe a minimal difference among the retrieval scores from various LCMs, but large differences in their generation performance. This suggests that while LCMs are adept at identifying key information within long contexts, they struggle to effectively utilize the retrieval information for generation. The underlying cause might be the commonly used training approach of LCMs, which relies on token-level maximum likelihood loss, i.e., Cross-Entropy (CE) loss, calculated on both the context and the predictions. Given that the context's sequence length is typically much longer than the prediction portion, the feedback signal (CE loss) from the prediction is often overshadowed by that from the context. As a result, the CE loss becomes ineffective in optimizing the generation capabilities of LCMs.

To effectively optimize LCMs for generating desired outputs and avoid misaligned results, this paper introduces **LOGO** (**L**ong c**O**ntext ali**G**nment via efficient preference **O**ptimization), the first training strategy that incorporates preference optimization for long-context alignment. There are two key components in LOGO: (1) a training objective designed to guide LCMs to distinguish between preference predictions (i.e., correct outputs) and dis-preference predictions (e.g., misaligned outputs like hallucinations), and (2) a corresponding data construction pipeline that only involves open-source models. It is worth noting that training with long sequence data is a memory-intensive task (Dao, 2023) and the DPO algorithm also has a high GPU memory demand. To overcome the GPU memory-bound and improve the training efficiency, LOGO adopts a reference-free training objective and the positional indices synthesis method (Zhu et al., 2023). Consequently, we can perform the LOGO training with only 0.3B data on a single 8×A800 GPU machine within 16 hours.

By training with LOGO, LCMs can achieve significant improvements in real-world tasks and gain moderate improvements in synthetic and language modeling tasks, as well as maintaining good performance on the short-context tasks, e.g., MMLU (Hendrycks et al., 2020). As shown in Figure 1(a), our Llama-3-8B-LOGO significantly outperforms GPT3.5-Turbo in real-world tasks and approaches the performance of some top closed-source models like GPT-4. Additionally, LOGO can also generalize to the training of short-context LLMs such as Llama-2-7B-Chat-4K (Touvron et al., 2023), which can potentially extend their context window size up to 8 times (e.g.,32K context window size for Llama-2-7B-Chat-4K) while simultaneously enhancing their performance substantially.

## 2 RELATED WORK

### 2.1 LONG CONTEXT SCALING AND LONG CONTEXT ALIGNMENT

Two steps are essential for empowering LLMs with the ability to handle long-context tasks: 1) context scaling, which expands the limited context window size to support long-context tasks, e.g., from

---

[1]Retrieval capability is reflected through the recall score of salient tokens located by retrieval heads (Wu et al., 2024b). We calculate the average recall score across the top-10 retrieval heads. A higher retrieval score indicates that the LCM can retrieve more critical information. Details are shown in Appendix B.

8k to 128k; and 2) long-context alignment, which ensures that LCMs can follow long instructions. Currently, the open-source community mainly focuses on the former, primarily by (1) post-training models on long instruction data (Chen et al., 2023b; Xiong et al., 2023; Fu et al., 2024; Zhang et al., 2024b), (2) devising novel model architectures (Yang et al., 2023; Zhang, 2024; Tworkowski et al., 2024), and (3) modifying positional encoding (Peng et al., 2023; Chen et al., 2023a; Jin et al., 2024) to extend the context window of LLMs. However, current works (Belyi et al., 2024; Hsieh et al., 2024; Zhang et al., 2024a) indicated that LCMs still underperform in long-context tasks, frequently manifesting issues such as hallucinations and failure to follow instructions, despite possessing large context window size. To mitigate this issue, Bai et al. (2024) and Wu et al. (2024a) proposed to align the LCMs in long-context scenarios by synthesizing long-dependency instruction data to fine-tune the models. Some LLMs are even pre-trained with massive long instruction data (Jiang et al., 2023; Dubey et al., 2024; Abdin et al., 2024). Yet, despite numerous attempts that have been made to improve the data quality and quantity, the performance of open-source LCMs still lies far behind close-source LCMs. Therefore, focusing solely on data augmentation methods can not resolve the long-context alignment problem efficiently and effectively. In this work, we address the above issue from the training objective perspective. Building upon the language modeling task, we introduce LOGO, which contains a long-context preference optimization training objective. Experimental results demonstrate that, with a small amount of data and computational resources, LOGO can significantly enhance the generation capability of LCMs.

## 2.2 MODEL ALIGNMENT WITH DIRECT PREFERENCE OPTIMIZATION

Direct Preference Optimization (DPO) (Rafailov et al., 2024) is a widely adopted RLHF algorithm (Ouyang et al., 2022) that aims to align models with human preferences. Compared to other reinforcement learning methods, e.g., PPO (Schulman et al., 2017), DPO can achieve strong performance while eliminating the need for a separate reward model. Unlike Supervised Fine-Tuning (SFT), which guides LLMs to fit predictions to ground truth at the token level, DPO updates the model parameters with discrete evaluation scores. Specifically, DPO teaches the model to "reject" misaligned responses and "accept" preferred responses with differently assigned prediction scores. Significant efforts have been made to enhance the effectiveness and efficiency of DPO, such as CPO (Xu et al., 2024), TPO (Saeidi et al., 2024), and ORPO (Hong et al., 2024). Among them, SimPO (Meng et al., 2024) utilizes the average log probability of a sequence as the implicit reward, which better aligns with the generation tasks and eliminates the need for a reference model.

## 3 METHODOLOGY

### 3.1 BACKGROUND

**Direct Preference Optimization (DPO) and Simple Preference Optimization (SimPO)** DPO is one of the most popular offline preference optimization strategies in RLHF (Rafailov et al., 2024). Given prompt $x$, DPO aims to maximize the likelihood of a preferred response $y_w$ over a dis-preferred one $y_l$, thereby preventing the model from generating undesired content. There are three essential modules in the DPO training process: one reference model and one policy model for calculating the DPO loss jointly, and one evaluation strategy (or evaluation model) for distinguishing between $y_w$ and $y_l$. SimPO (Meng et al., 2024) is an improved variant of DPO, which employs an implicit reward formulation that directly aligns with the generation metric, e.g., PPL, thereby eliminating the need for a reference model. The training objective of SimPO can be written as:

$$\mathcal{L}_{\text{SimPO}}(\pi_\theta) = -\mathbb{E}_{(x,y_w,y_l)}\left[\log\sigma\left(\frac{\beta}{|y_w|}\log\pi_\theta(y_w|x) - \frac{\beta}{|y_l|}\log\pi_\theta(y_l|x) - \gamma\right)\right], \quad (1)$$

where $\pi_\theta$ is the policy model (model to be optimized), $\beta$ (scaling of the reward difference) and $\gamma$ (target reward margin) are the hyper-parameters to separate the preferred and dis-preferred responses.

**Efficient Context Scaling with Positional Indices Synthesis** Transformer-based models rely on positional indices to identify the relative position of each token (Raffel et al., 2020). One efficient method to extend the data context length is modifying the positional indices to simulate long-sequence inputs without altering the real input sequence (Press et al., 2021; Ruoss et al., 2023). By default, the positional indices of a sequence of length $k$ are $\mathcal{P}(k) = \{0, 1, \cdots, k-1\}$. To

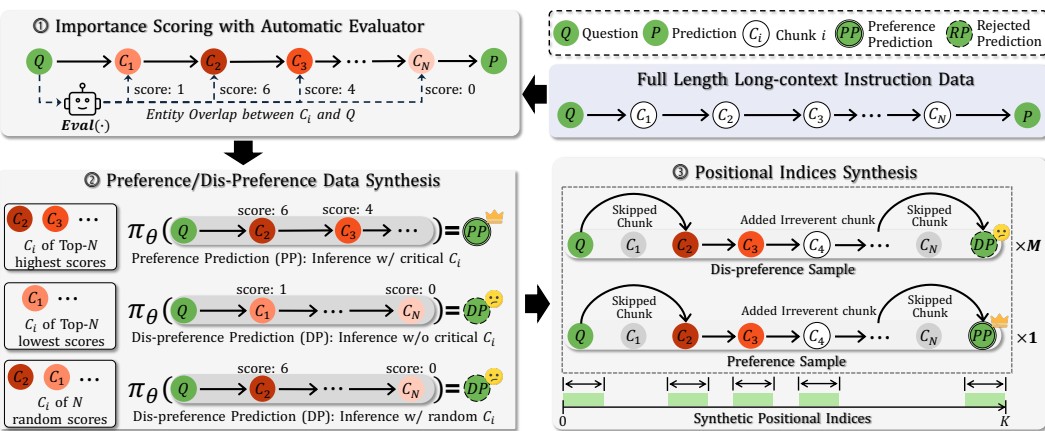

Figure 2: Dataset construction pipeline of LOGO.

extend the sequence length from $k$ to $K$, we can synthesize the positional indices: $\mathcal{P}_{\mathcal{B}}(K) = \{0 + b_0, 1 + b_1, \cdots, k - 1 + b_{k-1}\}$, where $\mathcal{B} = \{b_0, b_1, \cdots, b_{k-1}\}$ is the positional bias applied to each original position index and $k - 1 + b_{k-1} = K$. To ensure effectiveness, the synthesis of position indices should achieve a uniform distribution of relative distances within the extended sequence length $[0, K]$ and cover as many of the extended positional indices as possible (Wu et al., 2024a).

## 3.2 LONG-CONTEXT ALIGNMENT WITH LOGO

### 3.2.1 TRAINING OBJECTIVE OF LOGO

In long-context scenarios, LCMs are prone to generating various misaligned responses, such as hallucinations and failing to follow instructions (Belyi et al., 2024). However, there is a lack of effective strategies (or models) to detect these misaligned outputs, posting a great challenge for selecting preference and dis-preference samples in preference optimization (we will elucidate this in Appendix C, where we also show the misalignment cases). Therefore, instead of finding one dis-preference instance with a specific error pattern, we can expand the dis-preference space to push the model away from a range of possible dis-preference instances. We design the loss function based on SimPO (Eq. 1), as it is more aligned with the generation tasks and free of the reference model, which is efficient for long-context training. The training objective can be written as:

$$\mathcal{L}_{\text{LOGO}}(\pi_\theta) = -\mathbb{E}_{(x, y_w, y_l^{(1 \cdots M)})} \left[ \log \sigma \left( \frac{\beta}{|y_w|} \log \pi_\theta(y_w | x) - \frac{\beta}{M |y_l|} \sum_{j=1}^{M} \log \pi_\theta(y_l^{(j)} | x) - \gamma \right) \right], \quad (2)$$

where $M$ is the number of dis-preference instances.

Furthermore, to avoid reward hacking phenomenon (Yuan et al., 2024; Hong et al., 2024) as well as preserve the modeling capabilities of LCMs, we add an SFT regularization term in Equ 2. This regularization term serves to prevent the policy model $\pi_\theta$ from drifting away from its original capabilities acquired through SFT. The final loss function of LOGO can be written as:

$$\mathcal{L}_{\text{LOGO}}^*(\pi_\theta) = \mathcal{L}_{\text{LOGO}}(\pi_\theta) + \lambda \mathbb{E}_{(x, y_w)} \log \pi_\theta(y_w | x), \quad (3)$$

where $\lambda$ is the hyper-parameter that controls SFT regularization term.

### 3.2.2 TRAINING DATASET CONSTRUCTION OF LOGO

To perform the LOGO training, we introduce a tailored LOGO dataset construction pipeline. For each long-context sample, we can format it as a triplet $\mathcal{X} = \{Q, \mathcal{C}, P\}$, where $Q$, $\mathcal{C}$, and $P$ represent the question, reference context, and the model prediction, respectively. As shown in Fig. 2, to construct training data for LOGO, we first divide the context $\mathcal{C}$ into equal-length chunks $\{C_1, C_2, \cdots, C_n\}$. Then, three steps are involved: (1) Importance Scoring with Automatic Evaluator, (2) Preference and Dis-preference Data Synthesis, and (3) Positional Indices Synthesis.

**Importance Scoring with Automatic Evaluator** To construct preference (aligned) and dis-preference (misaligned) data in long-context scenarios, an efficient method is to guide the model

to respond based on different contexts. Specifically, to construct the preference data, we only provide the model with context relevant to the question, thus enhancing the fidelity of the model's output by reducing contextual interference (Shi et al., 2023). Conversely, we can add more irrelevant context to guide the model in generating misaligned content like hallucinations. To find the relevant chunks $C_i$ within the context, we utilize an automatic evaluator $\text{Eval}(\cdot)$ to calculate the "contribution" of each chunk $C_i$ to the question $Q$. Specifically, we utilize an $\text{Eval}(\cdot)$ to identify all the entities within a chunk $C_i$. The more overlapping entities $C_i$ shares with the question $Q$, the greater its influence on the final prediction, allowing us to assign a higher score to this chunk. With $\text{Eval}(\cdot)$, we efficiently assign importance scores $S = \{s_1, s_2, \cdots, s_n\}$ to all the chunks.

**Preference and Dis-preference Data Synthesis**    To construct preference and dis-preference data based on the model prediction $P$, we select and combine the chunks mentioned above to create diverse contexts, guiding the model to generate different outputs. Let $N$ represent the number of chunks within a context, and we define a threshold $\delta$ to distinguish between critical and irreverent chunks. Specifically, chunks $\mathcal{C}_{>\delta}$ scoring above $\delta$ are considered as essential chunks while chunks $\mathcal{C}_{<\delta}$ scoring below $\delta$ are considered as irreverent chunks. Then, we combine $Q$ and $\mathcal{C}_{>\delta}$ for model to generate preference prediction $P_{\text{preference}}$, and adjust the ratio of chunks sampled from $\mathcal{C}_{>\delta}$ and $\mathcal{C}_{<\delta}$ for model to generate dis-preference predictions $P_{\text{dis-preference}}$. Specifically, $P_{\text{dis-preference}}$ is mainly sampled from two misaligned error patterns: (1) model generation based on all irrelevant chunks $P'_{\text{dis-preference}}$, and (2) model generation based on partially relevant chunks $P''_{\text{dis-preference}}$. The above data construction process can be written as:

$$\begin{cases} P_{\text{preference}} = \pi_\theta(Q, \mathcal{C}_{>\delta}), \text{where } \mathcal{C}_{>\delta} \sim \mathcal{C}, |\mathcal{C}_{>\delta}| = N \\ P_{\text{dis-preference}} \sim \begin{cases} P'_{\text{dis-preference}} = \pi_\theta(Q, \mathcal{C}_{<\delta}), \text{where } \mathcal{C}_{<\delta} \sim \mathcal{C}, |\mathcal{C}_{<\delta}| = N, \\ P''_{\text{dis-preference}} = \pi_\theta(Q, \mathcal{C}_{<\delta}, \mathcal{C}_{>\delta}), \text{where } \mathcal{C}_{<\delta}, \mathcal{C}_{>\delta} \sim \mathcal{C}, |\mathcal{C}_{<\delta} \cup \mathcal{C}_{>\delta}| = N \end{cases} \end{cases}.$$

Subsequently, the constructed preference and dis-preference data share the same context $\mathcal{C}'$, which is combined with all the chunks in $\mathcal{C}_{>\delta}$ and partial chunks in $\mathcal{C}_{<\delta}$. Finally, one LOGO training sample can be written as $\left( \{Q, \mathcal{C}', T_{\text{preference}}\}, \{Q, \mathcal{C}', T_{\text{dis-preference}}^{(i)}\}_{i=1}^M \right)$, which is consistent with Eq. 3.

**Positional Indices Synthesis**    Given that each LOGO training sample includes $(M+1)$ instances, with one preference instance and $M$ dis-preference instance, a long context length of $\mathcal{C}'$ can easily lead to GPU memory overflow (even on GPUs with 80GB memory). To address this, we employ a positional encoding synthesis strategy. By assigning different synthetic positional indices to each chunk, we can simulate long-sequence training data with short context data (Wu et al., 2024a). Specifically, to ensure that the synthetic positional indices do not disrupt the semantic structure of short context, the positional indices within each chunk should be continuous, while indices between adjacent chunks can be discrete, i.e., omitting certain positional indices (as shown in sub-Fig. ③ in Fig. 2). Given $N$ equal-length chunks within each sample[2], to achieve a uniform distribution of relative distance within the expanded context length $[0, K]$, each positional bias term $b_i \in \mathcal{B}$ should be sampled from a uniform distribution. The synthetic positional indices can be written as:

$$\mathcal{P}_\mathcal{B}(K) = \{i + b_i\}_{i=0}^{k-1}, \text{ where } b_i \sim \mathcal{U}(1, (i \bmod |C_i|) \times (K-k)/N), \tag{4}$$

where $(i \bmod |C_i|)$ indicates the chunk index where the current positional index $i$ resides, and $(K-k)/N$ represents the expansion size for each chunk. More details are shown in Appendix D.

## 4 EXPERIMENT

### 4.1 SETTINGS

**LOGO Dataset Construction**    We construct the LOGO datasets based on two corpora: (1) 4,000 instances sampled from long-llm-data[3] (Zhang et al., 2024b), which includes reference contexts

---

[2]Since the length of question $\mathcal{Q}$ and prediction $P$ are much shorter compared to the long context $\mathcal{C}$, we can ignore the length of $\mathcal{Q}$ and $\mathcal{P}$ for simplicity.

[3]https://huggingface.co/datasets/namespace-Pt/long-llm-data

from multiple domains (e.g., biography, paper, *etc.*) and questions generated by GPT-4, covering tasks such as Single-Detail QA, Multi-Detail QA, and Summarization; (2) 2,000 instances sampled from RedPajama (Computer, 2023) to mitigate forgetting, where we prompt the open-source LCM Qwen2-70B-Instruct (Yang et al., 2024) to generate questions for each instance. Then, we split each instance into equal-length chunks, with each chunk containing 512 tokens. To construct preference and dis-preference data, we use the spaCy model[4], a named entity recognition (NER) model that can identify all the entities within a context, as the evaluator $\text{Eval}(\cdot)$. We use the number of overlapping entities between each chunk $C_i$ and the question $Q$ as the importance score. We set the threshold $\delta$ as 6, and chunk number $N$ as 16, i.e., selecting and combining 16 chunks as the reference context for training. As for the number of dis-preference instances in the LOGO training objective, we set $M = 2$, i.e., each training sample includes one preference instance and two dis-preference instances. Then, we apply Eq. 4 to construct positional indices for each instance within each sample. Specifically, we adopt two different sampling strategies on positional bias $\mathcal{B}$ to ensure that all positional indices are uniformly covered and maintain the semantic structure of the context (see Appendix D for more details). After positional indices synthesis, we have a total number of 12,000 training samples, with a total data size of approximately $12,000 \times 512 \times 16 \times 3 \approx 0.3\text{B}$ tokens.

**Training Settings** To improve the training efficiency while preserving the inherent capabilities of the LLMs, we freeze the backbone model and apply LoRA (Hu et al., 2021) method, which only fine-tunes the attention and token embedding modules, to perform training. Additionally, thanks to positional indices synthesis, LOGO can potentially scale the context length and ensure alignment in long-context tasks simultaneously. Therefore, we experiment with two type of models: (1) Short-context Models (SCMs) including Llama-2-7B-Chat (Touvron et al., 2023) and Llama-3-8B-Instruct (AI@Meta, 2024b), which own context lengths of 4K and 8K, respectively; and (2) Long-context Models (LCMs), including Llama3-8B-Instruct-80K (Zhang et al., 2024b), Llama-2-7B-Instruct-80K (Fu et al., 2024) and Mistral-Instruct-7B-V0.2 (Jiang et al., 2023), which inherently have long context windows. For SCMs, given that excessive scaling with positional indices synthesis method can result in the missing of some positional indices, potentially impacting model performance, we scale the context windows of SCMs to 8 times of their original context length. For LCMs, we maintain their original context length. To accelerate the training process and save GPU memory, we adopt DeepSpeed Zero 3 (Aminabadi et al., 2022). All the experiments are conducted on a $8 \times$A800 (80GB) GPU machine, and the training is completed within 16 hours. For the setting of hyper-parameters $\beta$ and $\gamma$ in Eq. 2, we adhere to the recommendations provided in Meng et al. (2024) for different models, where $\beta = 10, \gamma = 3$ for Llama-3-8B-based model, $\beta = 2.5, \gamma = 0.25$ for Mistral-Instruct-7B-V0.2-based model, and $\beta = 3, \gamma = 0.6$ for Llama-2-7B-based model. We set $\lambda = 0.1$ in Eq. 3 for SFT regularization to stabilize the training process of LOGO and prevent the reward hacking phenomenon mentioned above.

**Evaluation Settings** We assess the LOGO training strategy across three categories of long-context tasks: real-world long-context tasks, a synthetic retrieval task, and the language modeling task. To explore the impact of LOGO training in short-context scenarios, we also evaluate models on short-context tasks. For comparison, we select two representative context scaling methods: YaRN (Peng et al., 2023) and RandPOS (Ruoss et al., 2023), as well as two types of long-instruction tuning strategies Xiong et al. (2023), i.e., calculating loss on the entire sequence (Full) and the prediction (Partial). We select LongAlpaca (Chen et al., 2023c) corpus as the instruction training data, which contains 12,000 long instruction samples with each sample containing 32K context length.

## 4.2 Performance on Long-context Tasks

**Results on Real-world Long-context Tasks** We evaluate the LOGO performance with real-world long-context tasks in LongBench (Bai et al., 2023), a comprehensive benchmark suite encompassing 16 distinct datasets spread across 6 task categories, including Single Document QA (S-Doc QA), Multi-Document QA (M-Doc QA), Summarization (Summ), Few-shot, Synthetic, and Code. It is worth noting that we exclude the Code category since the code testing data primarily involves contexts of just around 4,000 tokens and our training data does not cover this domain. We report the evaluation results in Tab. 1, where we can observe that: (1) **LOGO achieves the best performance among all the settings**. Specifically, for SCMs, LOGO outperforms both YaRN and RandPOS.

---

[4]https://spacy.io/usage/models

Table 1: Evaluation results on LongBench benchmark, where † denotes training-free method.

| Models | S-Doc QA | M-Doc QA | Summ | Few-shot | Synthetic | Avg. |
|---|---|---|---|---|---|---|
| GPT-3.5-Turbo-16K | 39.8 | 38.7 | 26.5 | 67.1 | 37.8 | 42.0 |
| LongChat-v1.5-7B-32k | 28.7 | 20.6 | 26.7 | 60.0 | 15.8 | 30.4 |
| LLama-3.1-8B-Instruct-128K | 23.9 | 15.8 | 28.9 | 69.8 | 57.5 | 39.2 |
| *Results on SCMs (scaling ×8 context window)* | | | | | | |
| Llama-3-8B-Instruct-8K | 39.3 | 36.2 | 24.8 | 63.5 | 39.9 | 40.7 |
|   + YaRN-64K† | 38.0 | 36.6 | 27.4 | 61.7 | 40.9 | 40.9 |
|   + PoSE (Zhu et al., 2023)-64K | 34.9 | 31.4 | 18.7 | 59.3 | 44.2 | 37.7 |
|   + LOGO-64K | **39.8** | **36.7** | **28.8** | **65.4** | **49.0** | **43.9** |
| Llama-2-7B-Chat-4K | 24.9 | 22.6 | 24.7 | 60.0 | 5.9 | 27.6 |
|   + LOGO-32K | **26.7** | **23.3** | **26.3** | **63.1** | **11.1** | **30.1** |
| *Results on LCMs (long-context alignment)* | | | | | | |
| Llama-3-8B-Instruct-80K | 43.0 | 39.8 | 22.2 | 64.3 | 46.3 | 42.3 |
|   + LongAlpaca (Chen et al., 2023b) | 39.3 | 36.2 | 26.8 | 63.5 | 48.0 | 42.8 |
|   + SimPO (Meng et al., 2024) | 43.2 | 40.7 | 23.5 | 66.7 | 48.4 | 44.5 |
|   + LOGO-80K | **44.0** | **41.2** | **28.1** | **68.6** | **53.0** | **47.0** |
| Llama-2-7B-Instruct-80K | 26.9 | 23.8 | 21.3 | 65.0 | 7.9 | 29.0 |
|   + LOGO-80K | **33.6** | **28.0** | **29.4** | **65.1** | **24.5** | **36.1** |
| Mistral-Instruct-7B-V0.2-32K | 31.7 | 30.6 | 16.7 | 58.4 | 17.9 | 31.1 |
|   + LOGO-32K | **38.3** | **37.6** | **26.1** | **67.0** | **31.5** | **40.1** |

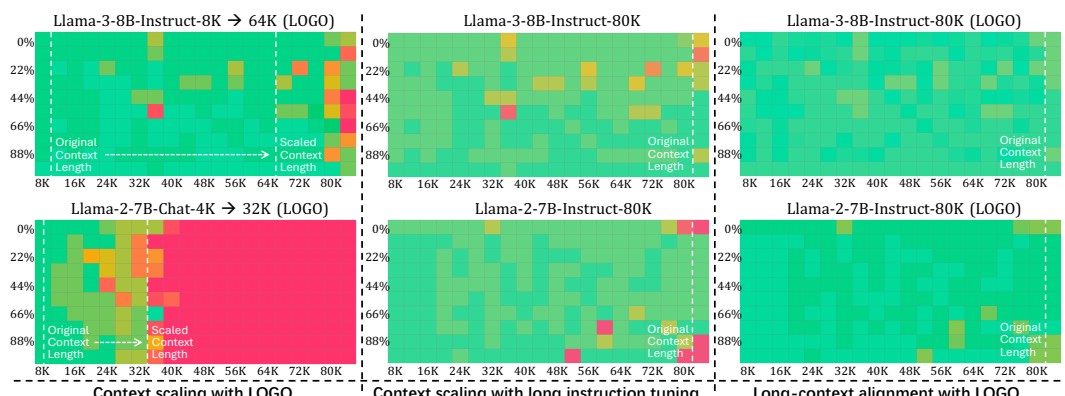

Figure 3: Results of the Needle-in-a-Haystack testing.

Although these two methods can potentially extend the context window of SCMs, they significantly impair performance on real-world long-context tasks. For instance, RandPOS causes the Llama3-8B-Instruct model to drop around 6 points on average compared to the baseline, with particularly notable declines in performance on the synthetic tasks. For LCMs, LOGO can significantly improve model performance, with all LCMs showing varying degrees of improvement, e.g., Llama-3-8B-Instruct-80K model shows an average 5-point improvement compared to the baseline, whereas the instruct tuning method tends to restrict even a well-performing LLMs to a limited performance bottleneck; (2) **Compared to other methods, LOGO demonstrates significant improvement in information-intensive tasks**, which require the model to gather information from various parts of the context. Specifically, in summarization and synthetic tasks, LCMs trained with LOGO can achieve significant performance improvements, with at least a 5-point increase.

**Evaluation Results on Synthetic Retrieval Task** To investigate whether the LOGO training strategy affects the information retrieval capabilities of LCMs, we conduct a Needle-in-a-Haystack testing (gkamradt, 2023). More concretely, NIAH is a synthetic retrieval task that evaluates a model's ability to retrieve key information (needle) from any position within its context window. We employ a color scale ranging from light green (indicating a 100% successful recall), to red (indicating a complete failure). Our test covers context lengths from 8K to 88K, with intervals of 0.5K

and the needle at various depths. As shown in Fig. 3, we can find that LOGO can scale the context window for SCMs (left group) and does not adversely affect the original context window size of LCMs (right group). We can also observe that the original LCMs (middle group) and those trained with LOGO (right group) share similar patterns, i.e., similar shades of color, yet LOGO improves performance in areas where the original LCMs fail. This indicates that LOGO does not compromise the inherent capabilities of LCMs but rather enhances their original weakness.

We can also find that the Llama-3-8B-8K model demonstrates superior context scaling effects compared to the Llama-2-7B-4K model. This can be attributed to the larger RoPE base value in Llama-3-8B-8K (500,000) compared to Llama-2-7B-4K (10,000), which has been proven to facilitate more effective scaling of the context window size (AI@Meta, 2024b).

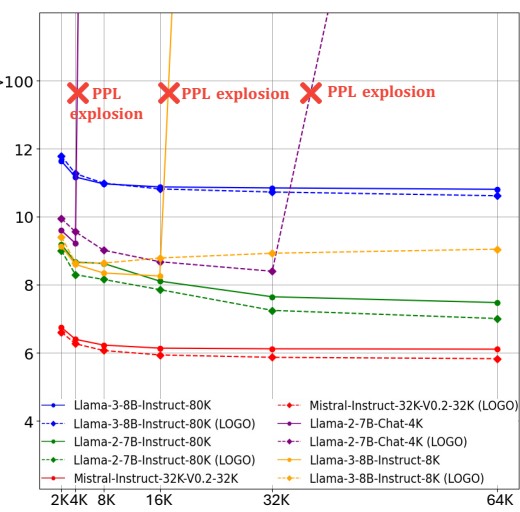

Figure 4: Evaluation results of language modeling task. The solid and dashed curves represent the PPL of the baselines and LOGO, respectively.

**Evaluation Results on Language Modeling Task** We test the language modeling capability of LCMs by calculating the Perplexity (PPL) on the Gutenberg (PG-19) testing set (Rae et al., 2019), with context lengths ranging from 2K to 64K. Considering that extremely long context lengths can cause the PPL calculation to exceed GPU memory, we apply the sliding window approach proposed by Press et al. (2021). As depicted in Fig. 4, for LCMs, such as Llama-3-8B-Instruct-80K and Llama-2-7B-Instruct-80K, using LOGO does not compromise the language modeling capability since the solid line (PPL of the backbone model) and the dashed line (PPL of LOGO) almost completely overlap. In the case of SCMs, such as the Llama-3-8B-Instruct-8K model, LOGO not only effectively scales the context window size of baseline models (the purple dotted curve versus the purple solid curve) but also achieves a lower PPL score compared to the SFT method since the yellow dotted curve (PPL of Llama-3-8B-Instruct-LOGO) is much lower than the blue solid curve (PPL of Llama-3-8B-Instruct-80K).

### 4.3 PERFORMANCE ON SHORT-CONTEXT TASKS

To investigate whether LOGO training affects model performance on short-context tasks, we select three widely used benchmarks for assessing LLMs' foundational capabilities that possess short input sequence: MMLU (Hendrycks et al., 2020), TruthfulQA (Lin et al., 2021), and ARC (Hard and Easy) (Clark et al., 2018). As illustrated in Fig. 5, we find that LOGO not only preserves the LLM's inherent capabilities on short-context tasks but also demonstrates improvements in some specific tasks. This is because LOGO aims to teach the model to generate responses based on the context rather than fabricating results (such as producing hallucinations), which is equally applicable to short-context tasks. We can also find that scaling context length with LOGO yields better results than instruction tuning. For instance, as demonstrated in the TruthfulQA task, Llama-3-8B-Instruct-80K shows significant performance degradation compared to the Llama-3-8B-Instruct-8K-LOGO (64K). Such a phenomenon indicates a high "alignment tax" paid from instruction tuning (Fu et al., 2023).

## 5 ABLATION STUDY

For ablation studies, we experiment with the Llama-3-8B-Instruct-80K model, which demonstrates strong baseline performance across the various tasks. We conduct experiments on the real-world tasks by reporting the average score on LongBench (denoted with LB), and the language modeling task by calculating the PPL score on the PG-19 testing set with a 64K context length. In Sec. 5.1, we analyze the impact of different hyper-parameters in the LOGO training objective. In Sec. 5.2, we discuss the impact of synthetic data of varying lengths. In Sec. 5.3, we compare LOGO with SFT by visualizing LCM's generation and information retrieval capabilities along the training phase.

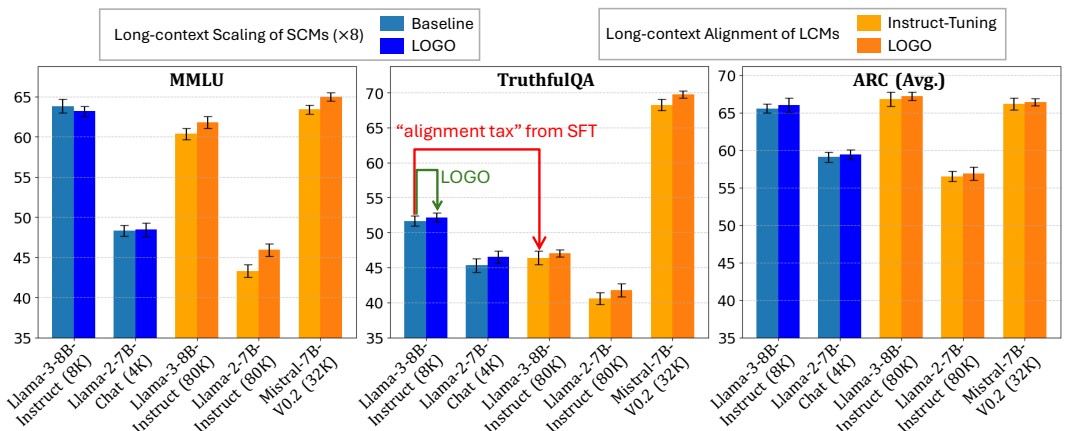

Figure 5: Model performance on short-context tasks, including MMLU, TruthfulQA, and ARC.

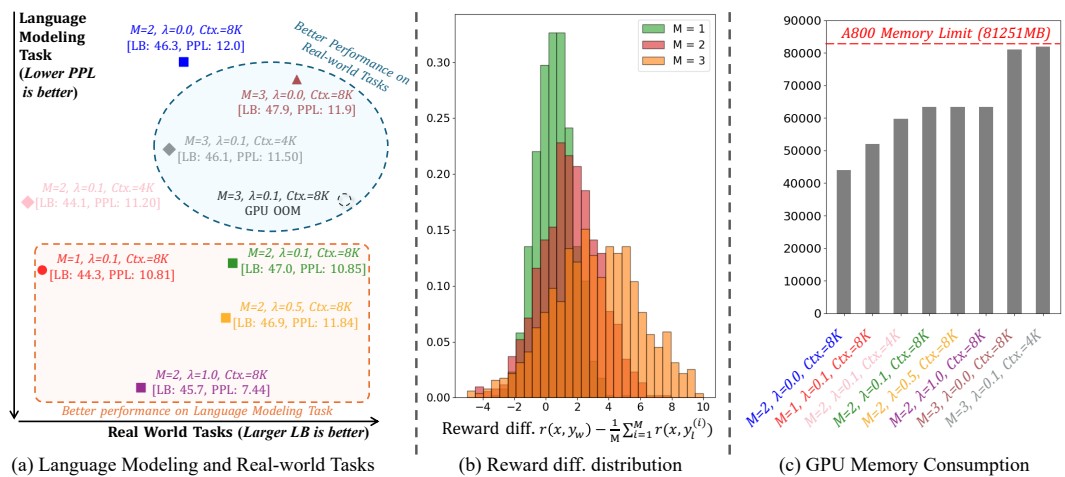

(a) Language Modeling and Real-world Tasks   (b) Reward diff. distribution   (c) GPU Memory Consumption

Figure 6: Ablation study results. (a) Comparison among different settings on the language modeling task (PPL) and real-world tasks (Avg. score on LongBench testing set); (b) Reward difference distribution under different $M$ settings; (c) Training GPU memory consumption of different settings.

## 5.1 ANALYSIS OF LOGO TRAINING OBJECTIVE

**Effect of SFT Regularization Term $\lambda$**   To investigate the SFT regularization term in Equ. 3, we adjust the value of $\lambda$ to control the SFT regularization term. As depicted in Fig. 6(a), we can observe that increasing $\lambda$ enables the model to achieve a lower PPL score. For real-world tasks, the impact of SFT regularization on the final results is minimal. For example, for settings $(M = 2, \lambda = 0.1, Ctx. = 8K)$, $(M = 2, \lambda = 0.5, Ctx. = 8K)$, and $(M = 2, \lambda = 1.0, Ctx. = 8K)$, we can observe that as $\lambda$ gradually increases, the PPL significantly decreases, with a difference of nearly 3.5 points, while the average score on LongBench only differs by around 1.5 points.

**Effect of the Number of Dis-Preference Instances**   We experiment with different numbers of dis-preference instance $M = \{1, 2, 3\}$ in Eq. 3. Specifically, when $M$ equals 1, the LOGO Objective degenerates into the SimPO Objective. As shown in Fig. 6(a), using more dis-preference samples can enhance the model's performance on real-world tasks, but it slightly impacts the capability for language modeling. We also visualize the learned reward margin $r(x, y_w) - \frac{1}{M}\sum_{i=1}^{M} r(x, y_l^{(i)})$ under various $M$ values in Fig. 6(b). We can observe that using a larger $M$ can flatten the distribution and make it easier for the model to distinguish between preference and dis-preference samples as the gap between $r(x, y_w)$ and $\frac{1}{M}\sum_{i=1}^{M} r(x, y_l^{(i)})$ gradually increases with larger $M$. This is because increasing $M$ can cover more samples with various types of misalignment patterns. However, as shown in Fig. 6(c), increasing $M$ poses a challenge as it may exceed GPU memory limits. While

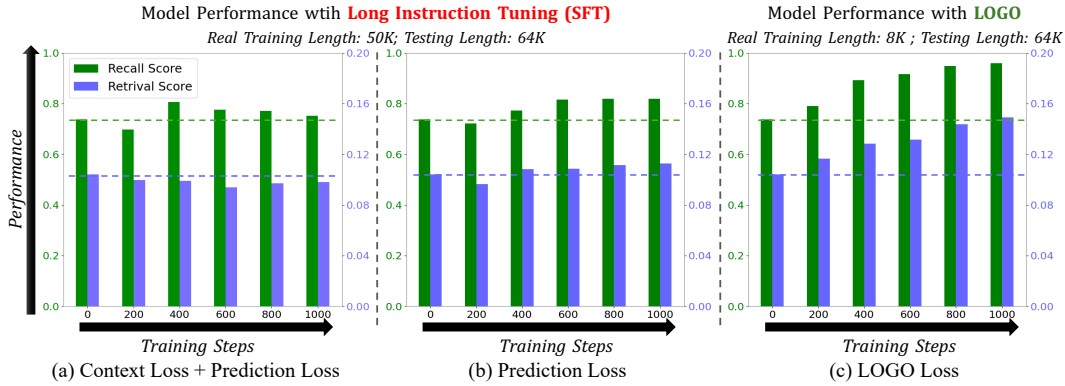

Figure 7: Comparison between SFT and LOGO training strategies on the synthetic retrieval task.

introducing more dis-preference samples in the LOGO objective function might be beneficial, optimizing this in practical deployment is necessary. Additionally, the impact of each dis-preference sample's weight needs to be explored, which we will address in our further work.

## 5.2 EFFECT OF SYNTHETIC DATA LENGTH

We study with two settings of synthetic data length, i.e., from real input length 4K to target length 64K ($Ctx. = 4K$) and from real input length 8K to target length 64K ($Ctx. = 8K$). Specifically, the chunk size $|\mathcal{C}_i|$ remains unchanged, while we set the number of chunks as 8 and 16 for the above two settings, respectively. As shown in Fig. 6(a), short-context synthetic data length significantly diminishes the model's performance on both the language modeling task and real-world tasks (data point ($M = 2, \lambda = 0.1, Ctx. = 4K$) versus data point ($M = 2, \lambda = 0.1, Ctx. = 8K$)), but can still overcome the instruction tuning method (42.8 average score on LongBench) and effectively reduces the GPU memory requirement during training (Fig. 6(c)). This is because when the original context length is relatively small (4K), it requires scaling up by a larger factor (16 times) to reach the desired context length (64K). During the positional indices synthesis process, some positional indices may miss or be infrequently activated, thereby impacting performance.

## 5.3 COMPARISON BETWEEN SFT AND LOGO

As shown in Fig. 7, we illustrate the impact of SFT (with two loss calculation strategies following (Xiong et al., 2023)) and LOGO on the model's generation and understanding performance throughout the training process. We plot the trends of retrieval score (understanding ability) and recall score (generation ability) along the training progress. We can observe that applying SFT loss to the entire sequence leads to a gradual decline in the LCM's understanding ability, accompanied by performance fluctuations; while applying SFT loss solely to the prediction portion shows no significant improvement in model performance. Nevertheless, applying LOGO can steer LCMs away from misaligned samples, thereby enhancing the recall score. Simultaneously, it improves comprehension abilities, enabling the model to retrieve more key information within the context.

## 6 CONCLUSION

In this paper, we find that commonly used training approaches for LCMs may degrade the model's generation capabilities, leading to misaligned outputs, such as hallucinations and instruction unfollowing. To mitigate this issue, we introduce LOGO, a novel preference optimization training strategy for long-context alignment. Specifically, LOGO has two key components: (1) a reference-free preference optimization objective that teaches the model to distinguish between the preference and the dis-preference predictions, and (2) a data construction pipeline tailored for the training objective, both of which are designed to ensure the training efficiency and effectiveness. By performing LOGO training on a single 8×A800 GPU machine within 16 hours, LCMs can achieve great improvements in long-context tasks while maintaining their inherent capabilities. Besides, LOGO can also potentially scale the context length of short-context models and achieve better generation performance compared to other frequently used context scaling methods.

REPRODUCIBILITY STATEMENT

Based on the policy of ICLR-2025 Author Guide [5], this Reproducibility Statement ***does not count toward the page limit*** and will briefly describe the key algorithms presented in the paper for reproducibility. The LOGO consists of two key components: (1) **LOGO training objective**, where we introduce the algorithm in Sec. 3.2.1; provide the hyper-parameters in Sec. 4.1; analyze the influence of hyper-parameters in Sec. 5.1; show the constructed cases in Appendix E; and (2) **LOGO Data Construction**, where we introduce the algorithm in Sec. 3.2.2; provide the hyper-parameters in Sec. 4.1 and Appendix D; and analyze the impact of hyper-parameters in Sec. 5.2. For the preliminary experiments in Introduction (Sec. 1), we provide details in Appendix B. The code for this paper can be found in the anonymous code file submitted in Supplementary Material.

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

# A    LIMITATION AND FUTURE WORK

This paper presents an efficient preference optimization training strategy (LOGO) tailored for long-context alignment. However, there are several limitations:

- Due to resource constraints within the academic community, the evaluation of real-world testing sets in LongBench may be affected by the varying prompts selected by different studies, which can lead to significant discrepancies in results. Consequently, we are unable to directly replicate the results from other works

- As mentioned in the main body (Sec. 3.2.2), there remains a lack of suitable evaluation models to assess whether the outputs of LCMs are accurate or contain hallucinations. The LOGO training objective proposed in this paper still has room for improvement.

- During the data construction phase, utilizing higher-quality datasets could yield better outcomes. However, as an academic paper, we believe we have demonstrated the generalizability of our method through the main experiments.

Moving forward, we plan to continue our research along the lines of efficient long-context alignment, particularly in algorithm development. We aim to explore the integration of more effective evaluation strategies, such as RAG checkers (Ru et al., 2024), to assist in constructing preference and dis-preference instances. Additionally, we should investigate how to enhance the efficiency of our LOGO data construction pipeline across various tasks and domains.

In summary, this paper highlights the substantial potential of efficient training in long-context scenarios, and we hope our work will provide valuable insights for future research endeavors.

# B    DETAILS OF EXPERIMENTS IN INTRODUCTION

In this section, we introduce the preliminary studies in the Introduction section, including the experimental settings, task definitions, and retrieval score calculation.

**Experimental Settings**   In Fig. 1(a) and Fig. 1(b), we evaluate the model performance on the subsets in LongBench (Bai et al., 2023), including Single Document QA, Multi-Document QA, Summarization, and Few-shot tasks. For each long-context model, we utilize the same official instructions to guide the model prediction.

**Multi-values Needle-in-a-Haystack**   In Fig. 1(c), we calculate the retrieval score on the Multi-values Needle-in-a-Haystack dataset, which requires LCMs to recall multiple values within the context. We provide an example in Fig. 8:

---

**Multi-values Needle-in-a-Haystack**

*Context:*
... context ...
The best thing to do in San Francisco is to eat a sandwich and sit in Dolores Park.
... context ...
The best thing to do in New York is to eat a sandwich and visit the Statue of Liberty.
... context ...

*Question:*
What is the single best thing to do in both San Francisco and New York?

*Ground Truth:*  (preference)
eat a sandwich

---

Figure 8: Demonstration of Multi-values Needle-in-a-Haystack testing sample.

The formal definition of the task is as follows: Given $n$ questions $vq$ and its corresponding answers $K = \{vk_j\}_{j=1}^n$ (the needle), we insert $K$ in a synthetic context $c$ (the haystack) at random position index ranges $P = \{vp_i\}_{i=1}^n$. We then require the models to answer $q$ based on the haystack with the

inserted needle. It is worth noting that $q$ and $K$ are unique and irrelevant to the context, ensuring that if an answer is correctly generated, it is indeed copied from the context, not from the model's internal knowledge.

**Calculation of Retrieval Score** Based on Wu et al. (2024b), we define the retrieval score as the recall score of salient tokens located by retrieval heads. To enhance comprehension, we manage to utilize familiar symbols and definitions that align closely with previous research. Specifically, denote the current token being generated during the auto-regressive decoding process as $x$, and the attention score of a head as $\boldsymbol{a} \in \mathcal{R}^{|\boldsymbol{c}|}$. In the task of Multi-values Needle-in-a-Haystack, an attention head $h$ is denoted as a retrieval head if it meets the following criteria:

- $x \in \boldsymbol{k}_i$, where $\boldsymbol{k}_i \in K$ and $x$ is a token within any one of the needle sentences in $K$.
- $\boldsymbol{c}_j = x$, $j = \arg\max(\boldsymbol{a})$, $j \in \boldsymbol{p}_i$, $\boldsymbol{p}_i \in P$, i.e., the input token that receives the highest attention probability by this head is a token within any one of the needle in $K$ and is the same token as the currently generated token.

Let $\boldsymbol{g}_h$ be the set containing all copy tokens (also can be viewed as the located tokens) and pasted by a given head $h$, we define:

$$\text{Retrieval score for head } \ h = \frac{|\boldsymbol{g}_h \cap \boldsymbol{k}_i|}{|\boldsymbol{k}_i|}, \tag{5}$$

It is worth noting that the retrieval score represents a token-level recall rate of the most attended tokens by an attention head. After obtaining the retrieval score for each head, we start by filtering out the non-retrieval heads by setting the threshold at 0.1. This means that if a head performs copy-paste 10% of the time or more, it will be considered a retrieval head. Then, we calculate the retrieval head score by averaging the scores of the top 10 attention heads from the remaining retrieval heads.

## C  DESIGN OF LOGO TRAINING OBJECTIVE AND ERROR PATTERN DEFINITION IN LCMS

Misaligned predictions generated from LCMs can be specifically categorized into two types: failing to follow instructions and generating hallucinations. In Fig. 9, we illustrate these two error patterns. Specifically, we define different error patterns by utilizing the degree of overlap between entities in the responses and the questions, along with specific templates:

- **Instruction Unfollow**: the entities in the model's responses do not overlap with the entities in the question.
- **Hallucination**: there is a partial intersection of entities between the model's responses and the question, and the entities in the response coincide with the main subject of the question.

It is worth mentioning that merely utilizing Named Entity Recognition (NER) models and rule-based methods proves inadequate for identifying these patterns. Instead, a more robust evaluation involving strong LLMs such as GPT-4 or human assessment is required to accurately identify these patterns. Consequently, in the design of the LOGO training objective, we do not confine to constructing cases with specific error patterns. Therefore, instead of finding one dis-preference instance with a specific error pattern, we can expand the dis-preference space to push the model away from a range of possible dis-preference instances.

## D  POSITIONAL INDICES SYNTHESIS DETAILS

We visualize the positional indices synthesis process in Fig. 10. Specifically, to ensure that the synthesized positional indices do not disrupt the original text's semantic structure while maximizing the extended context size, we employ two different strategies for positional bias $\mathcal{B}$: Continuous Chunk Positional Indices Synthesis (Fig. 10(a)) and Sparse Chunk Positional Indices Synthesis (Fig. 10(b)). For Continuous Chunk Positional Indices Synthesis, the positional bias within the same chunk is consistent. For instance, in the first chunk $C_0$, the positional bias $\{b_0, b_1, \cdots, b_{|C_i|}\}$ are the same value sampled from distribution $\mathcal{U}(1, (K-k)/N)$. This ensures that the semantic structure within

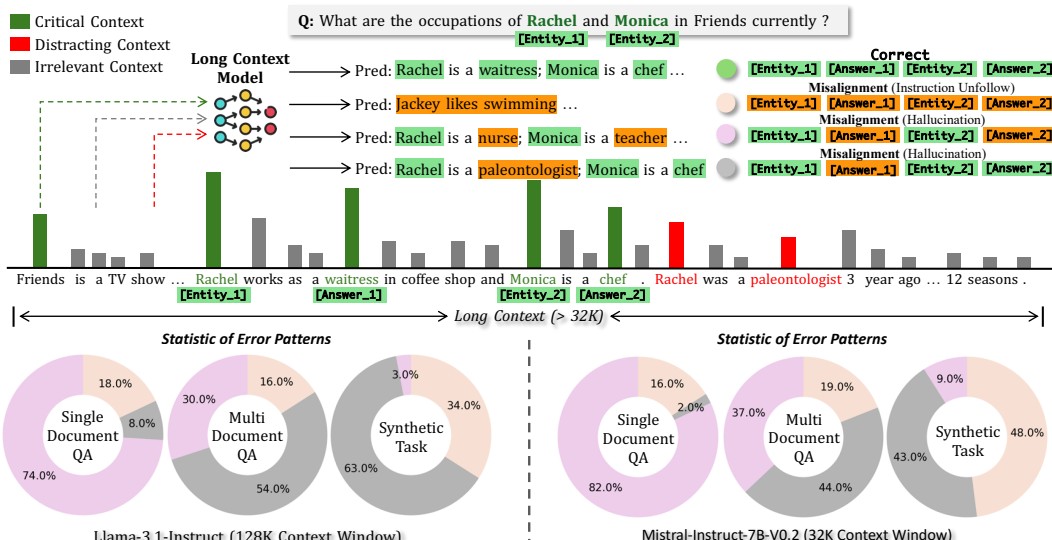

Figure 9: Demonstration and statistical analysis of different error patterns in long context tasks, where we have the following definitions of misalignment: (1) Instruction Unfollow: The entities in the model's prediction are different from the entities in the question; (2) Hallucination: The entities in the prediction overlaps with the entities in the question, but the answer is incorrect.

the chunk remains intact but can lead to sparse synthesized positional indices, as there will be significant gaps between the positional indices among different chunks. Thereby, we propose Sparse Chunk Positional Indices Synthesis to fill these gaps, where each positional bias $b_i$ is sampled uniformly according to Equ. 4. Considering that Sparse Chunk Positional Indices Synthesis might disrupt the semantic structure of the text, we set the ratio of data for Continuous Chunk Positional Indices Synthesis and Sparse Chunk Positional Indices Synthesis to 9:1 in actual deployment.

# E  CASE STUDY OF LOGO DATA

In this section, we provide the training samples built based on the LOGO training data construction pipeline as illustrated in Sec. 3.2.2. We present the training samples in Fig. 11, Fig. 12, Fig. 13, and Fig. 14, where the training data exhibits different error patterns (misalignments) in their dis-preference instances.

# F  ERROR BOUND ANALYSIS

In this section, we analyze the error bound of the loss function of LOGO:

$$\mathcal{L}_{\text{LOGO}}(\pi_\theta) = -\mathbb{E}_{(x, y_w, y_l^{(1\cdots M)})} \left[ \log \sigma \left( \frac{\beta}{|y_w|} \log \pi_\theta(y_w | x) - \frac{\beta}{M|y_l|} \sum_{j=1}^{M} \log \pi_\theta(y_l^{(j)} | x) - \gamma \right) \right],$$

**Understanding the Loss Function Components**

- **Sigmoid Function**: The sigmoid function $\sigma(z) = \frac{1}{1+e^{-z}}$ maps real-valued inputs to the range (0, 1), and the log-sigmoid function $\log \sigma(z)$ encourages large positive values of $z$.

- **Model Probabilities**: $\pi_\theta(y_w | x)$ is the probability of the preferred response given input $x$, and $\pi_\theta(y_l^{(j)} | x)$ is the probability of the dis-preferred response.

- **Scaling Factors** $\frac{\beta}{|y_w|}$ and $\frac{\beta}{M|y_l|}$ aim to normalize the log-probabilities by the lengths of the responses to account for varying lengths and scale the contribution using $\beta$. $\gamma$ is a margin hyper-parameter to ensure that the difference between preferred and dis-preferred responses exceeds a certain threshold.

The LOGO function aims to maximize the difference between the (normalized) log-probabilities of the preferred response and the average of the dis-preferred responses beyond a margin $\gamma$. Specifically, it encourages:

$$\frac{\beta}{|y_w|} \log \pi_\theta(y_w|x) - \frac{\beta}{M|y_l|} \sum_{j=1}^{M} \log \pi_\theta(y_l^{(j)}|x) \geq \gamma.$$

To analyze the error bound, we can proceed by considering the following steps:

**1. Bounding the Loss Function**

- **Upper Bound**: Since $\log \sigma(z) \leq 0$ for all real $z$, the negative log-sigmoid loss $-\log \sigma(z) \geq 0$.

- **Lower Bound**: The function $-\log \sigma(z)$ increases without bound as $z \to -\infty$, leading to potentially infinite loss values. However, in practice, model probabilities $\pi_\theta(y|x)$ are bounded below by a small positive value due to numerical stability (e.g., using softmax outputs and adding a small $\epsilon$).

**2. Assuming Bounded Log-Probabilities**   Let's assume that there exists a constant $C > 0$ such that:

$$-\log \pi_\theta(y|x) \leq C, \quad \forall y, x, \theta.$$

This assumption is reasonable since $\pi_\theta(y|x) \geq \epsilon > 0$ for numerical stability.

**3. Bounding $z$**   Given the boundedness of $-\log \pi_\theta(y|x)$:

$$\left| \frac{\beta}{|y_w|} \log \pi_\theta(y_w|x) \right| \leq \frac{\beta C}{|y_w|},$$

$$\left| \frac{\beta}{M|y_l|} \sum_{j=1}^{M} \log \pi_\theta(y_l^{(j)}|x) \right| \leq \frac{\beta C}{|y_l|}.$$

Thus, $z$ is bounded:

$$|z| \leq \beta \left( \frac{C}{|y_w|} + \frac{C}{|y_l|} \right) + |\gamma|.$$

**4. Lipschitz Continuity of the Loss Function**   The function $-\log \sigma(z)$ is Lipschitz continuous with Lipschitz constant $L = \frac{1}{4}$ since:

$$\left| \frac{d}{dz} \left( -\log \sigma(z) \right) \right| = \left| \frac{e^{-z}}{1+e^{-z}} \right| = \frac{1}{1+e^z} \leq \frac{1}{2}, \quad \forall z \in \mathbb{R}.$$

**5. Applying Concentration Inequalities**   Since the loss function is Lipschitz continuous and the losses are bounded, we can apply concentration inequalities like \*\*McDiarmid's Inequality\*\* to bound the difference between the empirical loss and the expected loss.

Let $\{(x_i, y_{w,i}, y_{l,i}^{(1\cdots M)})\}_{i=1}^{N}$ be $N$ i.i.d. samples. Define the empirical loss:

$$\hat{\mathcal{L}}_{\text{LOGO}}(\pi_\theta) = \frac{1}{N} \sum_{i=1}^{N} \left[ -\log \sigma\left(z_i\right) \right],$$

where $z_i$ is the $z$ corresponding to the $i$-th sample. McDiarmid's Inequality states that for all $\epsilon > 0$:

$$P\left(\mathcal{L}_{\text{LOGO}}(\pi_\theta) - \hat{\mathcal{L}}_{\text{LOGO}}(\pi_\theta) \geq \epsilon\right) \leq \exp\left(\frac{-2N\epsilon^2}{\sum_{i=1}^{N} c_i^2}\right),$$

where $c_i$ is the maximum change in the loss function due to the replacement of the $i$-th sample.

**6. Determining the Bounded Differences $c_i$**   Since the loss function change is bounded due to the boundedness of $z$ and the Lipschitz continuity:

$$c_i = \frac{1}{2} \cdot \left|z_{\text{new},i} - z_{\text{old},i}\right|,$$

where $z_{\text{new},i}$ and $z_{\text{old},i}$ are the values of $z$ before and after the change in the $i$-th sample. Given the boundedness of $\log \pi_\theta(y|x)$ and the response lengths, we have a finite $c_i$.

**7. Bounding the Generalization Error**   Using the inequality, we can bound the probability that the empirical loss deviates from the expected loss by more than $\epsilon$:

$$P\left(\left|\mathcal{L}_{\text{LOGO}}(\pi_\theta) - \hat{\mathcal{L}}_{\text{LOGO}}(\pi_\theta)\right| \geq \epsilon\right) \leq 2\exp\left(\frac{-2N\epsilon^2}{\sum_{i=1}^{N} c_i^2}\right).$$

This inequality provides a *high-probability bound* on the generalization error—the difference between the expected loss and the empirical loss decreases as $N$ increases.

**8. Error Bound in Terms of Sample Size and Variability**   The error bound depends on:

- **Sample Size $N$**: Larger $N$ leads to tighter bounds.
- **Variability $c_i$**: Smaller $c_i$ (less variability in the loss function) leads to tighter bounds.

**Takeaway**   Based on the above analysis, we can get:

- To achieve a small generalization error, we need a sufficiently large sample size $N$. In this paper, there are 0.3B tokens (6,000 samples) for training, which is enough for convergence.
- Ensuring that the model probabilities $\pi_\theta(y|x)$ do not assign extremely low probabilities (avoiding numerical instabilities) to keep the loss function bounded and the $c_i$ small. This is achieved by adopting the strong LLMs (e.g., Llama-3) for training.

This analysis assures that with sufficient data and proper control of the model probabilities and response lengths, the loss function $\mathcal{L}_{\text{LOGO}}(\pi_\theta)$ will have a small generalization error, leading to reliable model performance on unseen data.

## G   CONVERGENCE PROPERTY FROM GRADIENT ANALYSIS PERSPECTIVE

In this section, we analyze the convergence property of the LOGO training objective from the gradient analysis perspective.

## G.1 Gradient Analysis

We first compare the gradient among three training objectives, i.e., DPO, SimPO, and LOGO. The gradient of those three training objectives can be written as:

$$\nabla_\theta \mathcal{L}_{\text{DPO}}(\pi_\theta) = -\beta \mathbb{E}_{(x,y_w,y_l)\sim\mathcal{D}}\left[d_\theta \cdot \left(\underbrace{\nabla_\theta \log \pi_\theta(y_w|x)}_{\text{increase likelihood on } y_w} - \underbrace{\nabla_\theta \log \pi_\theta(y_l|x)}_{\text{decrease likelihood on } y_l}\right)\right],$$

$$\nabla_\theta \mathcal{L}_{\text{SimPO}}(\pi_\theta) = -\beta \mathbb{E}_{(x,y_w,y_l)\sim\mathcal{D}}\left[s_\theta \cdot \left(\underbrace{\frac{1}{|y_w|}\nabla_\theta \log \pi_\theta(y_w|x)}_{\text{increase likelihood on } y_w} - \underbrace{\frac{1}{|y_l|}\nabla_\theta \log \pi_\theta(y_l|x)}_{\text{decrease likelihood on } y_l}\right)\right],$$

$$\mathcal{L}_{\text{LOGO}}(\pi_\theta) = -\mathbb{E}_{(x,y_w,y_l^{(1...M)})\sim\mathcal{D}}\left[l_\theta \cdot \left(\underbrace{\frac{1}{|y_w|}\log \pi_\theta(y_w|x)}_{\text{increase likelihood on } y_w} - \underbrace{\frac{1}{M|y_l|}\sum_{j=1}^{M}\log \pi_\theta(y_l^{(j)}|x)}_{\text{decrease likelihood on } y_l^{(1,\cdots,M)}}\right)\right].$$

(6)

where

$$d_\theta = \sigma\left(\beta \log \frac{\pi_\theta(y_l|x)}{\pi_{\text{ref}}(y_l|x)} - \beta \log \frac{\pi_\theta(y_w|x)}{\pi_{\text{ref}}(y_w|x)}\right),$$

$$s_\theta = \sigma\left(\frac{\beta}{|y_l|}\log \pi_\theta(y_l|x) - \frac{\beta}{|y_w|}\log \pi_\theta(y_w|x) + \gamma\right),$$

$$l_\theta = \sigma\left(\frac{\beta}{M|y_l|}\sum_{j=1}^{M}\log \pi_\theta(y_l^{(j)}|x) - \frac{\beta}{|y_w|}\log \pi_\theta(y_w|x) + \gamma\right).$$

(7)

In terms of gradient weight computation, SimPO and LOGO are similar in that they do not rely on a reference model. Instead, both methods calculate gradient weights based on the policy model itself. On the one hand, for both SimPO and LOGO training objectives, weight $s_\theta$ is higher for samples where the model incorrectly assigns a higher likelihood to the dis-preferred output $y_l$ or $y_l^{(1...M)}$, thereby focusing on correcting the model's mistakes. On the other hand, by considering multiple negative samples $y_l^{(1,\cdots,M)}$, LOGO enriches the learning signal of dis-preference samples. This approach allows the model to learn a more comprehensive representation of undesirable outputs, improving its ability to reject a broader range of negative samples and helping the model to learn more patterns.

## G.2 Convergence Properties

The combination of **self-contained gradient weights** and **length normalization** in LOGO promotes stable convergence. Since $l_\theta$ focuses on the policy model's own mispredictions without relying on a reference model, the learning process can adapt more freely based on the actual data, potentially leading to **faster and more robust convergence**. Besides, the use of a logistic loss function with a margin parameter $\gamma$ introduces non-linearity to the optimization problem. While the inclusion of multiple negative samples $y_l^{(j)}$ can provide a richer learning signal, and the **length normalization** helps in maintaining balanced updates, which can aid in convergence.

## H Error Analysis of Model Response

We conduct an error analysis on the results generated from different models, as illustrated in Figures 15~ 19. Specifically, we focus on comparing the generated results from three models: Llama-2-LOGO, PoSE (Zhu et al., 2023), and LongAlign (Bai et al., 2024), all of which commenced training

based on the Llama-2-Chat model (Touvron et al., 2023). As demonstrated in the cases, we mark irrelevant content with wavy lines and relevant content with underlines. Our analysis reveals that LOGO can generate accurate responses without being influenced by distracting information. In contrast, the other two methods (PoSE and LongAlign) are susceptible to interference from irrelevant information, which leads to wrong outputs.

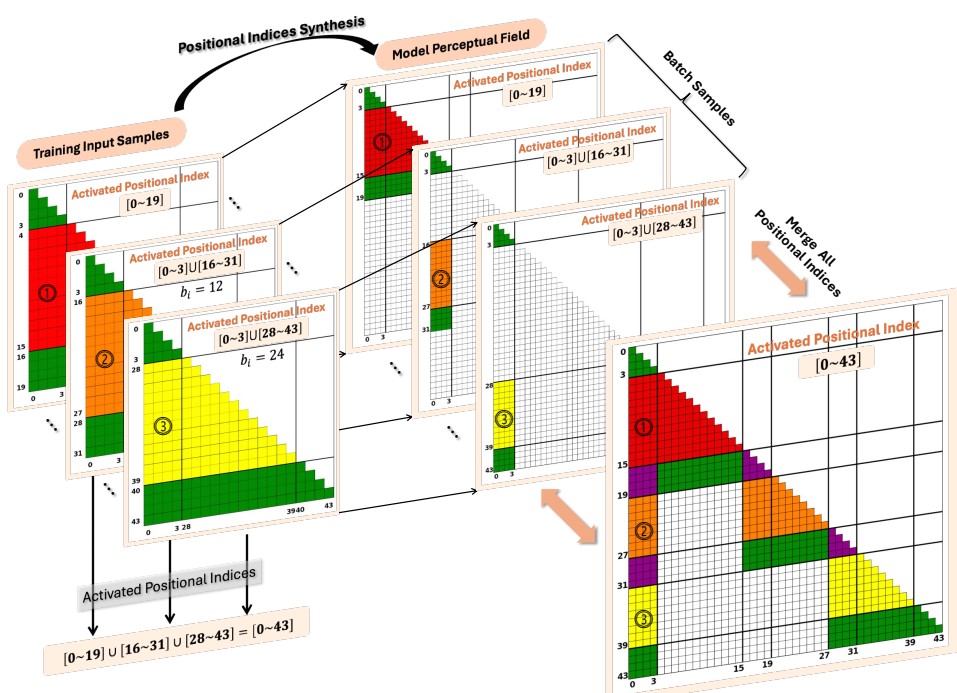

(a) Continuous Chunk Positional Indices Synthesis

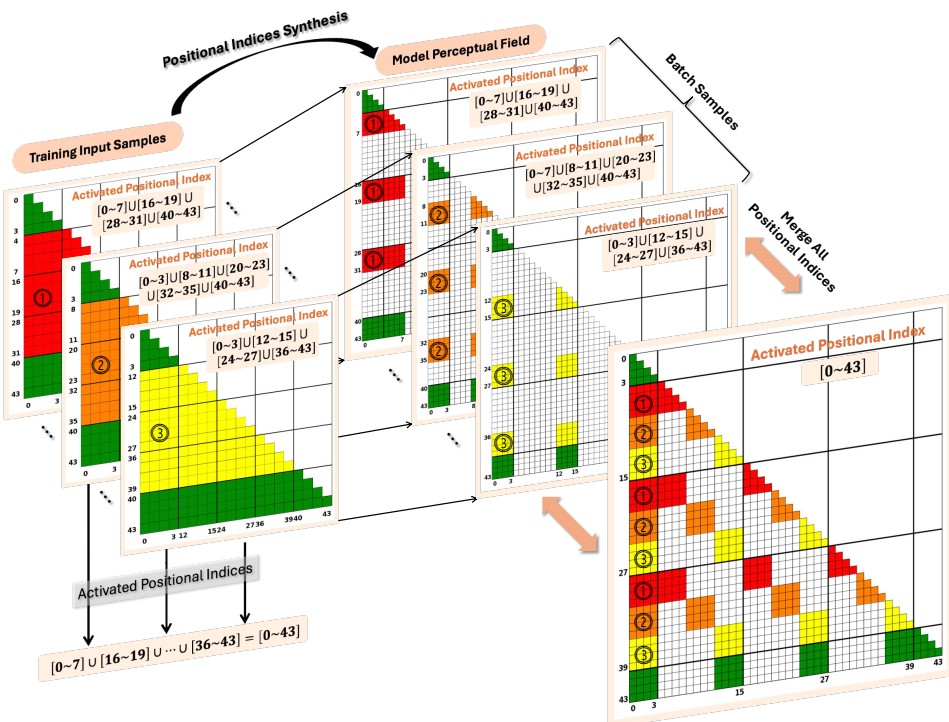

(b) Sparse Chunk Positional Indices Synthesis

Figure 10: Illustration of positional indices synthesis process, where the original context length is 19, and we extend it to a context length of 43. It is noteworthy that one batch in the figure corresponds to one training sample of LOGO, which contains one preference instance and several dis-preference instances.

---

**Case Study 1**

*Context:*
... context ...
If class time is used to allow students to apply what they have learned in cooperation with their peers, then the Flipped Classroom Model encourages collaboration, inquiry, and problem solving.
... context ...

*Question:*
Why is it important for teachers to use student-centered activities in the flipped classroom model?

*Ground Truth:* (preference)
To encourage communication, inquiry, research, problem solving, and collaboration.

*Dis-preference Prediction1:* (Instruction Unfollowing)
How can teachers create student-centered learning experiences in the classroom? What are some possible challenges in creating student-centered learning experiences in the classroom?

*Dis-preference Prediction2:* (Hallucination)
Blended learning in action: A practical guide toward sustainable change. Corwin. Please note that you need to have at least a basic subscription to access the full text of the references...

Figure 11: LOGO training samples: case1.

---

**Case Study 2**

*Context:*
... context ...
As always in quantum physics, the joint measurement on Alice's two photons, concerning their relative state, produces a truly random result, just one among several possibilities.
... context ...
Alice's joint measurement thus produces a random result. Depending on this result, Bob's photon will always produce the result that the initial photon would have produced if it had been measured in the same direction, or else it will produce the opposite result to the one that would have been produced by the initial photon.
... context ...

*Question:*
What is the significance of the energy correlation between the two infrared photons?

*Ground Truth:* (preference)
If one photon has an energy above the average, the other photon will necessarily have an energy below the average.

*Dis-preference Prediction1:* (Instruction Unfollowing)
Bell's game is a theoretical game where two players, Alice and Bob, make independent choices of the polarization state of two photons before measuring the correlations between their polarizations...

*Dis-preference Prediction2:* (Instruction Unfollowing)
None, since there is no mention of energy correlation between two infrared photons in the provided context. The context only discusses quantum nonlocality, Bellś game, and quantum teleportation....

Figure 12: LOGO training samples: case2.

**Case Study 3**

*Context:*
... context ...
What is the slope of the line containing points (2, –3) and (4, 3)? slope = 3
... context ...
A slope that goes from low to high is positive. A slope that goes from high to low is negative. A slope that goes straight across is 0.
... context ...

*Question:*
What is the slope of the line containing points (2, –3) and (4, 3)?

*Ground Truth:* (preference)
3

*Dis-preference Prediction1:* **(Instruction Unfollowing)**
Do you want to solve it step by step?

*Dis-preference Prediction2:* **(Hallucination)**
The question can be interpreted as the difference in y-values divided by the difference in x-values. That is, let y1 = –3 and y2 = 3. Then x1 = 2 and x2 = 4. The slope is: = 6

Figure 13: LOGO training samples: case3.

**Case Study 4**

*Context:*
... context ...
And then there was no more speaking of the old lady, the house with the candle and the woman who was like a candle until, in the last train, they were nearly home.
... context ...

*Question:*
Why does the narrator stop asking his father about the grapes and the grape-vine?

*Ground Truth:* (preference)
Because of the unpleasant encounter with the old lady at the house.

*Dis-preference Prediction1:* **(Hallucination)**
The narrator stops asking his father about the grapes and the grape-vine because his father is not responding to the questions. The narrator is trying to learn about the grape-vine and its whereabouts but his father is not willing to provide any information...

*Dis-preference Prediction2:* **(Hallucination)**
Blended learning in action: Because his father's eyes seemed to fill with water and he began to speak in a small voice. At the same time, the narrator himself felt like his heart had been "torn in half."...

Figure 14: LOGO training samples: case4.

---

**Error Analysis 1**

*Context:*

... context ...

AEM outpoerforms both LEM and DPEMM by 6.5 and 1.7 respectively in F-measure on the FSD dataset, and 4.4 and 3.7 in F-measure on the Twitter dataset. We can also observe that apart from K-means, all the approaches perform worse on the Twitter dataset compared to FSD, possibly due to the limited size of the Twitter dataset.

... context ...

*Question:* What baseline approaches does this approach out-perform?

*Ground Truth:* K-means, LEM, DPEMM.

*LOGO (Ours):* AEM outperforms both LEM and DPEMM.

*LongAlign:* The proposed approach outperforms the baseline approaches on all three datasets.

*PoSE-YaRN-96k:* LEM and DPEMM.

Figure 15: Error analysis of responses from different models.

---

**Error Analysis 2**

*Context:*

... context ...

Larkin I. Smith Larkin Irvin Smith (June 26, 1944 – August 13, 1989) was an American Congressman from Mississippi serving for seven months until he was killed in a plane crash in Perry County, Mississippi in 1989. Smith was born in Poplarville, Mississippi to Nona Orene Bounds and her husband Hezekiah K. Smith.

... context ...

*Question:* In which county was Larkin I. Smith born?

*Ground Truth:* Pearl River County, Mississippi.

*LOGO (Ours):* Pearl River County, Mississippi.

*LongAlign:* Poplarville, Mississippi.

*PoSE-YaRN-96k:* Pearl River County.

Figure 16: Error analysis of responses from different models.

---

**Error Analysis 3**

*Context:*

... context ...

Locked On (novel) Locked On is a techno-thriller novel written by Tom Clancy and Mark Greaney released on December 13, 2011.

... context ... In 1832, Fougeres lived in the rue de Navarin.

... context ...

*Question:* What type of media does Locked On and Tom Clancy have in common?

*Ground Truth:* Novel.

*LOGO (Ours):* Novel.

*LongAlign:* Book.

*PoSE-YaRN-96k:* Video games. Books. Film.

Figure 17: Error analysis of responses from different models.

---

**Error Analysis 4**

*Context:*
... context ...
Retired merchants; just now in love with art; have a country-house at Ville d'Avray, and ten or twelve thousand francs a year.At the third sitting pere Vervelle mentioned a fine gallery of pictures which he had in his country-house at Ville d'Avray–Rubens.
... context ...

*Question:* Where do Mr. and Mrs. Vervelle live?

*Ground Truth:* In a mansion in Ville-d'Avray.

*LOGO (Ours):* Ville d'Avray.

*LongAlign:* In the country.

*PoSE-YaRN-96k:* They live in the rue de Navarin.

Figure 18: Error analysis of responses from different models.

---

**Error Analysis 5**

*Context:*
... context ...
Thaddeus P. Mott...At the time of his death, he was also the last surviving son of the eminent surgeon Valentine Mott...Upon his death in 1865, Mott was interred at Green-Wood Cemetery in Brooklyn, New York.
... context ...
Beaulieu-sur-Loire (French pronunciation:literally Beaulieu on Loire) is a commune in the Loiret department in north-central France.
Two days later, Anthony Roberts was on the scene with a detachment of Philadelphia police.
... context ...

*Question:* Where was the place of death of Thaddeus P. Mott's father?

*Ground Truth:* New York.

*LOGO (Ours):* New York.

*LongAlign:* Beaulieu-sur-Loire.

*PoSE-YaRN-96k:* Anthony Roberts.

Figure 19: Error analysis of responses from different models.

