# OpenReview forum: "LOGO --- Long cOntext aliGnment via efficient preference Optimization"
_ICLR.cc/2025/Conference — Submitted to ICLR 2025_

### Official Review · Reviewer_D3My · 2024-10-27

**Soundness:** 2
**Presentation:** 3
**Contribution:** 3
**Rating:** 6
**Confidence:** 2

**Summary:**

This paper presents a novel preference alignment method for long-text models, combining position encoding expansion with human preference alignment techniques. For position encoding expansion, the authors propose splitting ultra-long contexts into multiple chunks, applying continuous position encoding within each chunk, and using a jump-based position encoding between chunks to achieve extended position encoding. In terms of preference alignment, the authors generate responses of varying quality by providing different qualities of context, treating higher-quality responses as preferred and lower-quality ones as non-preferred. These are then fed into SimPO for preference learning. Beyond SimPO loss, the authors also incorporate a weighted language modeling loss into the total loss.

Thanks to this unique position encoding expansion approach, the language modeling loss corresponding to strongly relevant contexts is not overly smoothed, thus improving optimization efficiency while reducing issues such as hallucination. On the other hand, the introduction of the powerful SimPO further strengthens the model’s instruction-following ability.

**Strengths:**

1. Innovative Training Strategy The introduction of LOGO, a long-context alignment method combined with preference optimization, improves the generation capabilities of LCMs.

2. Efficient Training LOGO adopts a position index synthesis method, allowing training to be completed with limited data and resources (8×A800 GPUs on a single machine in 16 hours), significantly improving training efficiency.

3. Significant Performance Improvement In real-world tasks, the Llama-3-8B-LOGO model significantly outperforms GPT-3.5-Turbo and approaches the performance of some top-tier closed-source models like GPT-4, while maintaining strong performance in short-context tasks as well.

**Weaknesses:**

More controlled experiments should be conducted, comparing the performance of models under the same experimental conditions: (1) using only instruction tuning, (2) using instruction tuning + SimPO (with SimPO’s positive and negative samples that already exist in the training corpus, rather than those generated by policy models or other LLMs), and (3) using the full LOGO method. These comparisons would clarify that the effectiveness of LOGO is not solely attributable to either instruction tuning alone or to the straightforward combination of instruction tuning and SimPO.

**Questions:**

1.	In the Preference and Dis-preference Data Synthesis section, you mentioned generating preferred data using πθ. Then, in the experimental section, you stated that you used long-llm-data as the training data. As far as I know, long-llm-data already includes standard answers. Did you generate additional answers using πθ beyond these standard answers? If so, what specific model was used as πθ—was it the policy model itself?

2.	You mentioned using long-llm-data as training corpus. To my understanding, this corpus, especially for the single-detail QA, multi-detail QA, and summarization datasets, was already instruction-tuning dataset. So, why do you mention at the end of the Evaluation Settings part that 12,000 data samples from LongAlpaca were used as instruction training data?

3.	Compared to using standard instruction tuning on long-llm-data, how much additional performance improvement does the SimPO loss provide? As far as I know, simple instruction tuning on long-llm-data already yields strong performance on LongBench.

---

> ### Author Response · Authors · 2024-11-16
> **Response to Reviewer D3My (Part I)**
>
> Thank you for acknowledging the novelty and efficiency of our work. We appreciate your concerns and suggestions, and we will address them one by one below.
>
> ---
>
> ### **Weakness 1 & Question 3**: Controlled Study of LOGO
> You are correct in suggesting that a controlled study of LOGO is necessary to fully understand its effectiveness. While our original paper included such studies, the results were scattered across Table 1 and Figure 6 (ablation study). For your convenience in the rebuttal, we refer you to the detailed comparison of the Llama-3 model in the **General Response**, as well as the detailed comparison of Llama-2 in our response to **Reviewer XuY2**. Below, we excerpt the conclusions for a clearer presentation:
>
> | Model                           | Type  | S-DocQA | M-DocQA | Summ  | Few-shot | Synthetic | Avg  |
> |---------------------------------|------|---------|---------|-------|----------|-----------|------|
> | Llama-2-7B-Chat-4K              | -    | 24.9    | 22.6    | 24.7  | 60.0     | 5.9       | 27.6 |
> | + Data Engineering              | SFT  | 26.9    | 23.8    | 21.3  | 65.0     | 7.9       | 29.0 |
> | + SimPO (80K)*                  | RL   | 29.1    | 24.2    | 25.7  | 64.4     | 16.3      | 31.9 |
> | + LOGO (80K)*                   | RL   | **33.6** | **28.0** | **29.4** | **65.1** | **24.5** | **36.1** |
>
> | Model                           | Type | S-DocQA | M-DocQA | Summ | Few-shot | Synthetic | Avg  |
> |---------------------------------|------|---------|---------|------|----------|-----------|------|
> | Llama-3-8B-Instruct-8K          | -    | 39.3    | 36.2    | 24.8 | 63.5     | 39.9      | 40.7 |
> | + Extending Ten-Fold Overnight  | SFT  | 43.0    | 39.8    | 22.2 | 64.3     | 46.3      | 42.3 |
> | + SimPO                         |      | 43.2    | 40.7    | 23.5 | 66.7     | 48.4      | 44.5 |
> | + LOGO                          |      | **44.0** | **41.2** | 28.1 | **68.6** | **53.0**  | **47.0** |
>
> It is evident that LOGO significantly enhances performance, while the improvements from SFT and SimPO have a clear upper limit. Simply combining SFT + SimPO (or other forms of DPO methods) has its limitations. The main reason is the absence of a suitable evaluation model to assess the quality of samples during the construction of preference data (which we also discussed with Reviewer XuY2). Additionally, in the long-context model generation process, various types of misalignments are difficult to judge with an evaluation model.
>
> By expanding the sampling space of negative samples, we can not only alleviate this issue but also allow the model to avoid more erroneous patterns. This is where LOGO differs significantly from DPO methods, as it addresses the challenge of constructing preference data in long-context scenarios where verification of results is extremely difficult.
>
> ---
>
> ### **Question 1**: About Training Data
> We understand your concerns and hope the following explanations can resolve your confusion:
>
> 1. **Use of Model πθ for Data Construction:**
>    The primary goal of our paper is to leverage the model's **inherent critical information retrieval capability**[3] to enhance its generative abilities. Therefore, we construct our training data starting from the model itself. The purpose of simulating critical/non-critical segments within the long context, as mentioned in our paper, is to simulate responses based on critical information vs. responses without using critical information. This is a progressive enhancement strategy that allows LCM to **avoid potential errors** it might make. Using other models to construct data would mean teaching πθ to learn from **other models' potential errors**, which deviates from our original intention. Therefore, we rely entirely on πθ to construct training data.
>
>
> 2. **Non-use of long-llm-data's Golden Data:**
> This is a critical point. The long-llm-data, originating from work [4], constructs golden data by providing GPT-4 with the long context. However, how can we ensure that GPT-4's generated data is correct (e.g., free from hallucinations)? While this work is promising, upon manual checking of some cases, we found significant issues. Below are our manual check results for 50 cases from each subset:
>
>     | Source  | Context has corresponding entities | Answer is correct |
>     |---------|:----------------------------------:|:------------------:|
>     | S-DocQA |                33/50               |        29/50       |
>     | M-DocQA |                30/50               |        29/50       |
>     | Book    |                38/50               |        29/50       |
>
> We can discover that **even GPT-4 makes numerous errors** and exhibits **significant hallucination**, producing content not present in the original context. In contrast, we need to construct data using the LOGO approach and we only provide critical/non-critical segments from the context to ensure the data quality.  That's why we use πθ to regenerate golden data.

---

> > ### Author Response · Authors · 2024-11-16
> > **Response to Reviewer D3My (Part II)**
> >
> > ### **Question 2**: Why Introducing LongAlpaca?
> > It seems there has been a misunderstanding. We introduced LongAlpaca solely to serve as an **SFT baseline for comparison** with our LOGO method. To clear up any confusion and to more clearly demonstrate the comparative effects, we have specified the use of LongAlpaca **in Table 1 of our revised manuscript**.
> >
> > ---
> >
> > ### **Reference**
> > [1] Fu, Yao, et al. "Data Engineering for Scaling Language Models to 128K Context." Forty-first International Conference on Machine Learning.
> >
> > [2] Zhang, Peitian, et al. "Extending Llama-3's Context Ten-Fold Overnight." arXiv preprint arXiv:2404.19553 (2024).
> >
> > [3] Wu, Wenhao, et al. "Retrieval head mechanistically explains long-context factuality." arXiv preprint arXiv:2404.15574 (2024).
> >
> > [4] Zhang, Peitian, et al. "Extending Llama-3's Context Ten-Fold Overnight." arXiv preprint arXiv:2404.19553 (2024).
> >
> > ---
> >
> >
> > We hope these explanations can solve your concerns and issues. You may refer to our revised manuscript for updated content. Additionally, we believe that LOGO represents a significant contribution to the field of long-context alignment. We hope you can reconsider our work, and an increase in the score would be greatly appreciated.

---

> > > ### Comment · Reviewer_D3My · 2024-11-25
> > > **Rebuttal Feedback**
> > >
> > > Thanks for the authors' efforts in addressing my concerns.
> > >
> > > I think this is a border paper. I maintain my score to show my accpetance tendency, but will not be surprised if this paper gets rejected.

---

### Official Review · Reviewer_XuY2 · 2024-10-30

**Soundness:** 3
**Presentation:** 2
**Contribution:** 1
**Rating:** 3
**Confidence:** 4

**Summary:**

This paper introduces LOGO, a novel training strategy that addresses the challenge of improving long context language models' (LCMs) generation capabilities while maintaining efficiency. While existing LCMs can effectively locate important information in long contexts, they often struggle with generating appropriate responses, leading to hallucinations and misaligned outputs. LOGO tackles this through a reference-free preference optimization approach that teaches models to distinguish between preferred and dis-preferred outputs, combined with an efficient data construction pipeline utilizing positional indices synthesis. The method's key advantage is its resource efficiency - requiring only 0.3B tokens of training data and 16 hours on a single 8×A800 GPU machine - while achieving comparable performance to GPT-4 on long-context tasks and maintaining performance on traditional benchmarks. The authors demonstrate LOGO's effectiveness across various tasks and its ability to extend context windows of existing models while enhancing their generation quality.

**Strengths:**

Combining preference optimization with long-context alignment addresses a gap in current LCM training methods.
Develops a creative data construction pipeline that effectively creates preference/dis-preference pairs without requiring extensive human annotation
Clear experimental methodology with detailed ablation studies that validate design choices
Well-structured presentation with clear problem motivation and solution development

**Weaknesses:**

Lack of rigorous evaluation methods for detecting misaligned outputs and hallucinations, which affects the quality assessment of preference/dis-preference pairs
While the paper provides implementation details, the quality of training data could significantly impact results, and the paper uses relatively simple datasets
The theoretical justification for why preference optimization works better than traditional methods in long-context scenarios could be stronger

**Questions:**

How does LOGO compare with recent baselines such as [1], and methods included in your related work?
Please add comparsion with pipeline using long context and preference optimization, for example LongRoPE[2]&SimPO[3].
Since your contribution focus on long context alignment, please eval it on corresponding benchmark, LongAlign[3].
Could you provide more theoretical analysis such as error bounds for LOGO and analyze its convergence properties?

[1] Zhao, Hao, et al. "Long Is More for Alignment: A Simple but Tough-to-Beat Baseline for Instruction Fine-Tuning." Forty-first International Conference on Machine Learning.
[2] Ding, Yiran, et al. "LongRoPE: Extending LLM Context Window Beyond 2 Million Tokens." Forty-first International Conference on Machine Learning.
[3]Meng, Yu, Mengzhou Xia, and Danqi Chen. "Simpo: Simple preference optimization with a reference-free reward." arXiv preprint arXiv:2405.14734 (2024).
[4] Bai, Yushi, et al. "Longalign: A recipe for long context alignment of large language models." arXiv preprint arXiv:2401.18058 (2024).

---

> ### Author Response · Authors · 2024-11-16
> **Response to Reviewer XuY2 (Part I)**
>
> Thank you for acknowledging the contribution of our work in addressing a gap in current Long Context Model (LCM) training methods. Below are our responses:
>
> ---
>
>
> ### **Question 1**: Lack of Rigorous Evaluation Methods for Detecting Misaligned Outputs and Hallucination
> You have raised a very astute observation regarding the evaluation of misaligned outputs and hallucinations in our dataset.
> We appreciate your insight and believe some misunderstandings need to be clarified by starting a discussion about ''*the challenges we faced in constructing our dataset and evaluating these misaligned outputs*''.
>
> As you rightly pointed out, we have written in our paper (**Lines 188-189**) that '' there is a lack of effective strategies (or models) to detect these misaligned outputs''.
> While it is theoretically feasible to evaluate certain types of misalignment, such as using manual evaluation or leveraging GPT-4 to assess the quality of synthetic data, the variety of misalignment types greatly complicates the evaluation process.
> Therefore, we aimed to minimize our dependence on evaluation models when constructing our data.
>
> LOGO addresses this issue from three aspects:
>
> 1) **Data Construction**: During data construction, we utilized non-critical segments within the context to guide the model to generate completely incorrect answers (dis-preference sample) and critical segments to guide the model to generate correct answers (preference sample). Here, the accuracy of preference samples is significantly higher than that of dis-preference samples since the responses are generated from the critical segments. Additionally, since we employed the RL method for training, our goal was to keep the model away from incorrect samples, rather than fitting to correct samples. Therefore, ensuring that the preference samples are acceptable and of higher quality than the dis-preference samples was sufficient, and we clearly achieved this.
>
> 2) **Objective Function of LOGO**: We expanded the space of negative samples in LOGO's objective function, allowing the model to avoid not just one type of error. This means that as long as the negative samples are generally incorrect, the training is effective.
>
> 3) **Utilizing the intrinsic knowledge of the model to serve our objectives.**: LOGO's goal is to leverage the long-context model's inherent critical information retrieval capabilities[1] (as mentioned in the Introduction section) and use the retrieval key information to generate a response. Therefore, regardless of whether there is an evaluation function, constructing preference data based on critical/non-critical segments within the context is reasonable because we aim to stimulate stronger generation capability based upon the inherent retrieval capability. It is worth noting that you may notice that stronger models (e.g., Llama-3) benefit more from LOGO training. This is because, during data construction, stronger LCMs can create higher-quality preference data (because those models have stronger capabilities), thereby further enhancing the effectiveness of training with synthetic data.
>
> From a deployability and scalability perspective, LOGO's data construction method and training objective function are not only **straightforward but also highly effective**.
>
> We acknowledge that the evaluation model is a crucial component in the synthesis of long-text data. However, any research is progressive, and LOGO has already demonstrated the significant role of negative(dis-preference) samples in preference alignment within the long-context field. We believe this is not just an issue for us alone, but rather a challenge that the entire long-context community should address collectively.
>
> ---
>
> ### **Question 2**: Comparison with more baselines
> We appreciate your suggestions and recognize the value of the references you provided, which primarily focus on training longer and better LCMs.
> It seems that in the early stages of LCM development, there was not a clear distinction between Context Window Scaling and Long-context Alignment, which is why we did not compare LOGO with these works directly due to the different settings.
>
> You also mentioned a pipeline approach, which we understand is similar to our setup: first, using a method (e.g., LongRoPE) to extend the model's context window size, and then using another strategy (e.g., SimPO) for long-context alignment. In fact, the experiments in the **"Results on LCMs" group of Table 1** in our paper employ the same method. Therefore, the comparison that needs to be added is between LOGO and SimPO on a long-context model.

---

> ### Author Response · Authors · 2024-11-16
> **Response to Reviewer XuY2 (Part II)**
>
> ### **Question 2**: Comparison with more baselines
>
> Below are the model results on the LongBench testing set. Considering that the baselines you mentioned are mainly conducted on the Llama-2 model, we chose Llama2 as the backbone model for additional experiments in this rebuttal. Specifically, we utilize Data Engineering [2] to scale the context window size. During the evaluation stage, we truncate the evaluation data length to the model's maximum context window size. Note that to ensure a fair comparison, we have listed the type and purpose of the dataset (Symbols: Context Window Scaling -> CWS, and Long-context Alignment -> LA, *denotes training from a long-context model [2]).
>
>
> | Model                           | Type | Purpose  | S-DocQA | M-DocQA | Summ  | Few-shot | Synthetic | Avg  |
> |---------------------------------|------|----------|---------|---------|-------|----------|-----------|------|
> | Llama-2-7B-Chat-4K              | -    | -        | 24.9    | 22.6    | 24.7  | 60.0     | 5.9       | 27.6 |
> | + Data Engineering ([2])        | SFT  | CWS       | 26.9    | 23.8    | 21.3  | 65.0     | 7.9       | 29.0 |
> | + PoSE (96K)                    | SFT  | CWS       | 26.5    | 26.0    | 11.9  | 55.0     | 5.5       | 25.0 |
> | + LongAlign (64K)               | SFT  | CWS + LA  | 28.3    | 26.4    | 24.4  | 64.3     | 7.1       | 30.1 |
> | + Refined-Alpaca-1k (70K)       | SFT  | CWS + LA  | 27.8    | 25.9    | 23.6  | 62.6     | 6.4       | 29.3 |
> | + LongAlpaca (80K)*             | SFT  | LA       | 25.3    | 22.8    | 25.9  | 61.2     | 10.5      | 29.1 |
> | + SimPO (80K)*                  | RL   | LA       | 29.1    | 24.2    | 25.7  | 64.4     | 16.3      | 31.9 |
> | + LOGO (80K)*                   | RL   | LA       | **33.6**    | **28.0**    | **29.4**  | **65.1**     | **24.5**      | **36.1** |
>
> We can observe that under different settings, **LOGO consistently performs the best**. We have also studied the related work you provided, among which LongAlign[4] and Refined-Alpaca-1k[5] both directly construct high-quality long-context data, achieving both context window scaling and alignment.
>
> Compared to these studies, LOGO differs significantly in terms of training data scale and experimental setups (one starting from short-context models, the other from long-context models). We believe that this comparison may **not be entirely fair**. However, we have included this type of work in the above table for your reference, and you will find that LOGO still performs better.
>
> For more experimental results on Llama-3 and other models, you can refer to the results shown in the **General Response** and Table 1 in our paper.
>
> ---
>
> ### **Question 3**: More Theoretical Analysis such as Error Bounds for LOGO and Convergence Properties
> Thank you for your inquiry regarding the theoretical underpinnings of LOGO, while I don't come from a traditional Machine Learning background, I have endeavored to provide as detailed a theoretical analysis as possible in the revised version of the manuscript based on my understanding, specifically in **Appendix F and G of the revised manuscript**.
>
> Since the numerous mathematical formulas make it impractical to present them fully in OpenReview, we put our conclusion here:
>
> From a theoretical standpoint, the error bound of LOGO depends on two key factors:
>
> - the volume of training data should sufficient (we have 0.3B tokens during the training stage);
> - the model probabilities πθ(y|x) not assign an unduly low probability. This aligns with our data construction process, which is entirely model-dependent (i.e., the training data is generated by the model itself), ensuring that the predicted probabilities are not set too low.
>
> ---
>
> ### **Reference**:
> [1] Wu, Wenhao, et al. "Retrieval head mechanistically explains long-context factuality." arXiv preprint arXiv:2404.15574 (2024).
>
> [2] Fu, Yao, et al. "Data engineering for scaling language models to 128k context." arXiv preprint arXiv:2402.10171 (2024).
>
> [3] https://huggingface.co/yaofu/llama-2-7b-80k
>
> [4] Bai, Yushi, et al. "Longalign: A recipe for long context alignment of large language models." arXiv preprint arXiv:2401.18058 (2024).
>
> [5] Zhao, Hao, et al. "Long is more for alignment: A simple but tough-to-beat baseline for instruction fine-tuning." arXiv preprint arXiv:2402.04833 (2024).
>
> ---
>
> We hope our responses and perspectives have addressed your concerns and misunderstandings. If you have any additional concerns or questions, please don't hesitate to let us know. We believe LOGO represents a significant advancement in the long-context alignment field due to its effectiveness and efficiency. We hope you can reconsider your evaluation of our work and an improvement in your score would be greatly appreciated, thank you.

---

### Official Review · Reviewer_WeBZ · 2024-11-03

**Soundness:** 3
**Presentation:** 2
**Contribution:** 3
**Rating:** 6
**Confidence:** 4

**Summary:**

This paper presents a novel approach to long-context language modeling, leveraging a combination of attention mechanisms and position encoding to improve performance on long-range dependencies. The method shows promising results in improving long-context understanding while maintaining computational efficiency.

**Strengths:**

-The paper proposes a new attention mechanism that combines the strengths of existing methods. This results in improved performance on long-range dependencies. It also enables efficient handling of long-context training with limited computational resources.

-The authors thoroughly evaluate their method on multiple benchmark datasets. They demonstrate its effectiveness in various settings and show a clear improvement over baseline methods.

**Weaknesses:**

-The core idea of using preference optimization for long-context alignment seems like a straightforward extension of existing methods such as DPO and SLiC. The position synthesis method shows similarities to existing techniques like ALiBi and RoPE. The paper's main contribution appears incremental rather than transformative.

-The preference optimization objective (Equation 3) is similar to DPO without significant modification. The position synthesis method lacks theoretical justification for its effectiveness. The training procedure fails to address the fundamental challenges of long-context understanding.

-Experimental Limitations: While the authors compare their method to several existing approaches, the comparison is not exhaustive, and some relevant methods are not considered.

**Questions:**

1.How does LOGO fundamentally differ from DPO in handling long-context scenarios?

2.What theoretical guarantees can be provided for the position synthesis method?

3.How does the method scale with increasing context lengths beyond 32k tokens?

4.Can you provide detailed analysis of failure cases?

---

> ### Author Response · Authors · 2024-11-16
> **Response to Reviewer WeBZ (Part I)**
>
> Thank you for your feedback and for acknowledging the computational efficiency of our method. Regarding the concerns and issues you raised, we would like to address and clarify these points in detail below.
>
> ---
>
> ### **Weakness 1 & 2**: Paper's Main Contribution Appears Incremental Rather Than Transformative & Fail to Address Fundamental Challenges of Long-Context Understanding
>
> We appreciate your feedback. However, we contend that the core argument of our paper centers on utilizing the inherent critical information retrieval capabilities of long-context models to improve their alignment effectiveness.
> We identify **three primary challenges for long-context models**: 1) the expansion of the context window, which is a key focus of current research; 2) the capability to retrieve essential information; and 3) the ability to process and generate responses based on this retrieved key information. Much of the existing research on aligning long-context models [1][2] is dedicated to the creation of higher-quality long-context data for fine-tuning (SFT) the model.
> Our perspective, however, highlights that models already possess robust capabilities for retrieving key information within a long context, a point indirectly confirmed by work [3]. This paper primarily explores how to leverage this retrieval ability to address misalignment issues in LCMs, such as hallucinations, which are vital for comprehension tasks.
>
> The methodologies we have adopted are not arbitrary combinations but rather carefully selected approaches designed to **harness the model's inherent capabilities**. For example, SimPO, when compared to traditional SFT, not only prevents misalignment but also demonstrates particular effectiveness in generation tasks. Similarly, PoSE (validated by studies [4][5]) presents a training strategy that is especially accessible to academic researchers who often face resource constraints, such as limited GPU availability.
>
> We firmly believe that the primary contribution of this work extends beyond merely combining existing methods. Rather, it lies in identifying an elegant yet powerful approach to activate the model's latent capabilities. Indeed, we would argue that the innovative application of **simple and effective** existing methods to address challenges in the long-context domain represents a significant contribution to the field. After all, isn't the clever utilization of established methods to solve complex problems a valuable advancement in itself?
>
> ---
>
> ### **Weakness 3**: Experimental Limitations:
> We acknowledge your concerns and would like to address them by providing additional experimental results to demonstrate the effectiveness of our approach:
>
> - We would respectfully direct your attention to the Llama-3 results presented in the **General Response**, as well as the specific findings related to the Llama-2 model detailed in our response to **Reviewer XuY2**. These comprehensive results offer valuable insights into our method's performance across various settings and in comparison with different baselines.
>
> - We recognize that direct comparisons may not be **completely equivalent** due to **variations** in datasets and training resources (as different baselines serve distinct purposes, which we elaborated on in our General Response - various methods enhance different aspects of LCMs, while our primary focus is on **improving long-context alignment capabilities**).
> Nevertheless, these additional comparisons provide substantial evidence supporting the effectiveness of our methodology.
>
> ---
>
> ### **Question 1**: How Does LOGO Fundamentally Differ from DPO?
>
> In our paper, we explicitly mention in lines 192-193 that ''we design the loss function based on SimPO''. Additionally, lines 149-150 clarify that ''SimPO is a variant of DPO''. Therefore, both LOGO and DPO aim to address the lack of dis-preference data during the SFT process. From this perspective, *most current work essentially aligns with the core principles of DPO*.
>
> However, the key distinction between LOGO and DPO (which was previously primarily suited for short-context tasks) lies in the complexity of constructing preference data and their application scenarios, particularly as evaluating model outputs in long-context scenarios presents significant challenges. To address these challenges, we have **expanded the space for dis-preference samples** in the LOGO objective function, rather than directly applying SimPO's loss function. Furthermore, we have incorporated an **SFT regularization term** to maintain the model's language modeling capabilities.
>
> Those modifications represent the fundamental difference from DPO.

---

> ### Author Response · Authors · 2024-11-16
> **Response to Reviewer WeBZ (Part II)**
>
> ### **Question 2**: Theoretical Guarantees for Position Synthesis Method
>
> We understand that the question of the theoretical guarantees for the position synthesis method is equivalent to understanding why Relative Position Encodings (RPE) are effective. For the effectiveness and theoretical proof of RPE, you can refer to the paper "Self-Attention with Relative Position Representations" available at https://arxiv.org/pdf/1803.02155. Other works such as [4][5] also use position synthesis to improve training efficiency.
>
> Simply put, for the Transformer model, the model learns a function $f(i, j)$ to understand the relative position information between any two tokens. The extrapolation ability of position encoding can be understood as the model's desire to learn information about larger $(i-j)$ values. Referring to the mathematical formula of RoPE, the attention formula can be written as:
> $$Att_{i,j} = \frac{(QK^{T} + f(i, j))V}{d^{1/2}}$$
> where $QK^{T}$ and $V$ are the query, key, and value, respectively. To perform length extrapolation, one only needs to consider how to make  $(i-j)$ larger. The conventional approach is to increase the sequence length to enlarge the value of  $(i-j)$, with each token corresponding to an absolute $i$ and $j$ value. As for position synthesis, it only needs to consider changing the values of  $i$ and $j$ without altering the actual sequence length.
>
> A potential issue is that many position indices may be missing in the positional synthesis method, and we illustrate how we compensate for the missing relative positions in **Appendix D** in our paper.
>
> ---
>
> ### **Question 3**: How LOGO Works When Increasing Context Lengths Beyond 32K
>
> Indeed, as sequence lengths increase, the primary consideration is how to use position encoding synthesis to cover more positions, which requires more short-context data to fill in the gaps. However, if the sequence length truly exceeds 32K, strategies like Ring Attention are necessary, utilizing multiple GPUs to share the training memory load. Below is the evaluation results when increasing context length to 32K:
>
> | Model                           | S-DocQA | M-DocQA | Summ | Few-shot | Synthetic | Avg  |
> |---------------------------------|---------|---------|------|----------|-----------|------|
> | Llama-3-8B-Instruct-80K         | 43.0    | 39.8    | 22.2 | 64.3     | 46.3      | 42.3 |
> | + LOGO (Sequence Length: 8K)    | 44.0    | 41.2    | 28.1 | 68.6     | 53.0      | 47.0 |
> | + LOGO (Sequence Length: 32K)   | **45.3**| **43.4**|**30.3**|**69.6** | **54.2**  |**48.6**|
>
> Specifically, based upon the original 8K context, we extended the context up to 32K by filling the context with irrelevant text (sampling from pre-trained corpus), allowing the model to further learn how to utilize key information from the context to generate responses. This approach not only tests the model's ability to handle longer contexts but also its capability to focus on relevant information amidst potentially distracting content.
> We observe that LOGO continues to demonstrate potential for further performance improvements (LOGO-8K improves from 42.3 to 47.0, while LOGO-32K improves from 42.3 to 48.6), which can be attributed to both more comprehensive positional encoding coverage and the benefits gained from increased computational resources (such as Ring attention).
>
> ---
>
> ### **Question 4**: Detailed Analysis of Failure Cases
> This is a valuable suggestion. Of course, we provide two failure cases here, and we have also supplemented the corresponding results in **Appendix H of the revised paper (Figure 15 ~ Figure 19)**.
>
> Below, we present two error cases:
>
> ```bash
> #### Context ####
> Thaddeus P. Mott ...
> [... context ...]
> At the time of his death, he was also the last surviving son of the eminent surgeon Valentine Mott...Upon his death in 1865, Mott was interred at Green-Wood Cemetery in Brooklyn, New York.
> [... context ...]
> Beaulieu-sur-Loire (French pronunciation: literally Beaulieu on Loire) is a commune in the Loiret department in north-central France.
> Two days later, Anthony Roberts was on the scene with a detachment of Philadelphia police.
> [... context ...]
>
> #### Question ####
> Where was the place of death of Thaddeus P. Mott’s father?
>
> #### Answer ####
> | Model            | Answer                |
> |------------------|-----------------------|
> | Ground Truth     | New York.             |
> | LOGO (Ours)      | New York.             |
> | LongAlign        | Beaulieu-sur-Loire.   |
> | PoSE-YaRN-96k    | Anthony Roberts.      |
> ```

---

> ### Author Response · Authors · 2024-11-16
> **Response to Reviewer WeBZ (Part III)**
>
> ```bash
> #### Context ####
> [... context ...]
> AEM outpoerforms both LEM and DPEMM by 6.5 and 1.7 respectively in F-measure on the FSD dataset, and 4.4 and 3.7 in F-measure on the Twitter dataset.
> [... context ...]
> We can also observe that apart from K-means, all the approaches perform worse on the Twitter dataset compared to FSD, possibly due to the limited size of the Twitter dataset.
> [... context ...]
> #### Question ####
> What baseline approaches does this approach out-bperform?
>
> #### Answer ####
> | Model            | Answer                                                |
> |------------------|-------------------------------------------------------|
> | Ground Truth     | K-means, LEM, DPEMM.                                  |
> | LOGO (Ours)      | AEM outperforms both LEM and DPEMM.                   |
> | LongAlign        | The proposed approach outperforms the baseline        |
> |                  | approaches on all three datasets.                     |
> | PoSE-YaRN-96k    | LEM and DPEMM.                                        |
> ```
> In these two cases where the context contains interfering information, we find that the answers generated by LOGO are consistent with the Ground Truth, indicating that LOGO can accurately retrieve relevant information in the context and utilize it for response.
> However, the other two methods (LongAlign and PoSE) are affected by irrelevant information within the context, leading to outputs that contain hallucinated or distorted information.
>
> ---
>
> ### **Reference**
> [1] Bai, Yushi, et al. "Longalign: A recipe for long context alignment of large language models." arXiv preprint arXiv:2401.18058 (2024).
>
> [2] Gao, Chaochen, et al. "Quest: Query-centric Data Synthesis Approach for Long-context Scaling of Large Language Model." arXiv preprint arXiv:2405.19846 (2024).
>
> [3] Wu, Wenhao, et al. "Retrieval head mechanistically explains long-context factuality." arXiv preprint arXiv:2404.15574 (2024).
>
> [4] Zhu, Dawei, et al. "PoSE: Efficient Context Window Extension of LLMs via Positional Skip-wise Training." The Twelfth International Conference on Learning Representations.
>
> [5] Wu, Wenhao, et al. "Long context alignment with short instructions and synthesized positions." arXiv preprint arXiv:2405.03939 (2024).
>
> ---
>
> We hope that our responses and perspectives can address your concerns and misunderstandings. If you have more concerns or questions, we would also appreciate hearing from you. We believe that LOGO represents a significant advancement in the field of long-context alignment due to its effectiveness and efficiency. Additionally, we hope you can reassess our work and an improvement in your score would be greatly appreciated, thank you.

---

### Official Review · Reviewer_Bjw9 · 2024-11-04

**Soundness:** 4
**Presentation:** 4
**Contribution:** 4
**Rating:** 6
**Confidence:** 5

**Summary:**

The paper introduces LOGO, a preference optimization training strategy to improve long-context alignment in language models. LOGO uses a reference-free preference objective and a position synthesis method to address memory constraints and efficiently train LCMs. With only 0.3B tokens on limited GPUs, LOGO achieves notable performance comparable to GPT-4 on real-world long-context tasks while preserving other model capabilities.

**Strengths:**

1. This work is the first to study long-context alignment. The topic and methods are both novel.

2. LOGO can extend the context window of short-context models, allowing for flexible adaptation across various LCM architectures.

3. Experiments on various benchmarks including needle in the hay-stack is promising.

4. The LOGO strategy effectively optimizes LCMs using limited data and resources, achieving comparable results with larger models.

**Weaknesses:**

1. If we use flash-attention (ring-attention) & deepspeed zero3 cpu offload, it is all right to train Llama-3-8B on 80k context (I already tested it). I think this should be a baseline to compare with the proposed Positional Indices Synthesis. The comparison should include both GPU memory, training hours and accuracy.

2. Would you please try longer context for evaluation? It seems that the longest context is commonly 80k in the paper, which might not be enough this year. For example, qwen2 models is commonly pre-trained as 128k context. It is able to train about 256k context with ring-attention (and the proposed Positional Indices Synthesis).

**Questions:**

What is the potential for scaling LOGO to models trained on diverse, multi-modal data? For example, long video VLM. I know that this might be hard to resolve in the rebuttal. This is just a discussion.

---

> ### Author Response · Authors · 2024-11-16
> **Response to Reviewer Bjw9 (Part I)**
>
> Thank you for your recognition of our work. We are pleased that you highly regard the contributions and soundness of our work.
> Below are our responses and supplementary experimental results addressing your concerns.
>
> ---
>
> ### **Question 1**: Analysis of GPU memory, training, and accuracy between preference optimization (LOGO) and traditional SFT
>
> We have conducted a detailed analysis and comparison of different training strategies, as follows:
>
> | Training Strategy | Configuration | GPU Memory Usage | Bsz per GPU | Total Throughput (8 GPUs) | Actual Training Length | Training Time (2000 steps) |
> |------------------|---------------|-----------|:-------------------:|-------------------------|----------------|---------------------------|
> | LOGO | No ring attention | 64GB | 3 | 24 samples | 12K | 16 hours |
> | SFT | No ring attention, DeepSpeed Zero3 | 79GB | 1 | 8 samples | 64K | 14 hours |
> | SFT + Ring Attention | Ring attention, DeepSpeed Zero3, Ring size=2 | 45GB | 1/2 | 4 samples | 64K | >24 hours* |
>
> *Note: The longer training time for SFT + Ring Attention is attributed to our PCIE network infrastructure.*
> - The GPU performance analysis related to LOGO's hyperparameters is presented in Figure 6(c) in the paper.
> - Performance comparisons can be found in Table 1, Figure 4, Figure 5, and Figure 7, as well as the additional experimental results in General Response.
> - Ring attention configuration involves context parallel communication between every two GPUs.
> - All experiments were conducted with the same number of training steps (2000).
>
> ---
>
> ### **Question 2**: Conducting experiments with longer context lengths, such as 128K and 256K
>
> You are correct in stating that 128K is now considered the threshold for long-context models. For context lengths of 128K, we can directly use YaRN's method to expand from 80K to 128K, and the model performance impact is not significant. For even longer context lengths, we need to extend the position synthesis method to a greater range and use more data to compensate for the context length gaps. We used 16K actual input (we still build data upon 8K critical/non-critical segments, and then fill the context with irrelevant text upon reaching the 16K context length) and simulated a 256K context length using position encoding synthesis, starting training from the Llama-3-8B-Instruct-LOGO-80K model. Here are the results on the LongBench testing set:
>
> | Model | S-DocQA | M-DocQA | Summ | Few-shot | Synthetic | Avg |
> |-------|----------|----------|-------|-----------|------------|------|
> | Llama-3-8B-Instruct-LOGO-80K | 44.0 | 41.2 | 28.1 | 68.6 | 53.0 | 47.0 |
> | Llama-3-8B-Instruct-LOGO-128K | 43.8 | 40.9 | 28.0 | 68.6 | 52.6 | 46.8 |
> | Llama-3-8B-Instruct-LOGO-256K | **44.9** | **42.6** | **29.8** | **69.4** | **53.9** | **48.1** |
> | Yi-6B-200K* | 39.1 | 25.1 | 33.8 | 25.6 | 56.6 | 36.0 |
>
> It can be observed that LOGO performs well under the 128K and 256K settings, with the 256K training scenario significantly outperforming the Yi-200K model. This improvement may be due to the introduction of noise in our training dataset (filling the length to 16K with irrelevant context sampled from pre-trained corpus), which further enhances the model's ability to locate information and utilize key information for responses. This also demonstrates the scalability of the LOGO method, which can be extended to even longer lengths.

---

> ### Author Response · Authors · 2024-11-16
> **Response to Reviewer Bjw9 (Part II)**
>
> ### **3. Discussion**: Scalability of LOGO to models trained on diverse, multi-modal data
>
> Thank you for your thought-provoking question about the potential for scaling LOGO to models trained on diverse, multi-modal data, such as long video VLMs. It's an exciting area of research, and coincidentally, I have recently been working on video generation. Here, I can provide an example from video generation to illustrate.
>
> **Controllability in Long Video Generation:**
> As you may have noticed, there are some significant controllability issues in long video generation, akin to the hallucination problem in text generation. Models may start following a prompt accurately but eventually deviate, introducing concepts and entities not present in the original prompt.
>
> **Keyframes and Controllability:**
> In the video domain, keyframes serve as pivotal points that define the structure and content of a video sequence. These keyframes can be thought of as the video equivalents of **critical information within the context** as mentioned in our paper. The concept of keyframes offers a promising avenue for enhancing controllability in video generation.
>
> **Harnessing Keyframes for Control:**
> By constructing keyframes that act as anchors for the video content, we can potentially guide the model to generate content that remains aligned with the initial prompt. This approach is analogous to how LOGO uses key information within a text context to generate responses. By creating hallucination keyframes, we can train the model to recognize and avoid deviating into unrelated content, similar to how LOGO steers the model away from non-preferred outputs.
>
> **Training Approach Inspired by LOGO:**
> Inspired by LOGO, we could train the model to generate videos that are more controllable and less prone to hallucination. This could involve:
>
> 1. **Preference Optimization for Videos:** Extending the preference optimization approach to video generation, where the model learns to differentiate between preferred (on-topic, coherent) and non-preferred (off-topic, incoherent) video content based on keyframes.
>
> 2. **Relative Position Representations for Video Context:** Adapting the relative position representations used in LOGO to understand the temporal relationships between keyframes and other video segments, helping the model to maintain coherence over longer sequences.
>
> We appreciate your interest in the potential applications of LOGO beyond text, and we believe that the future community will not only focus on whether we can create long-video (or other modalities) models but also on making that generation more controllable.
>
> ---
>
> We hope our responses have addressed your concerns. If you have more questions, please feel free to ask. We believe that LOGO will benefit the field and community of long-context alignment because it is a simple, scalable, and high-performing method. If possible, we kindly ask for a reassessment of our work, and an improvement in your score would be greatly appreciated.

---

> > ### Comment · Reviewer_Bjw9 · 2024-11-25
> > **Feedbacks on the rebuttal**
> >
> > Thanks for your detailed experiments and response. I think the results are meaningful and I will vote for acceptance and keep my original rate for this paper.

---

### Author Response · Authors · 2024-11-16
**General Response (Part I)**

First of all, thanks to all the reviewers for their thoughtful and constructive feedback. We deeply appreciate the time and effort each reviewer dedicated to reviewing our paper. We noticed several common questions and misunderstandings, which we will clarify in this **General Response**.

---

### **1. Motivation Behind LOGO and Baseline Selection**
We acknowledge the concerns raised by **Reviewer WeBZ** and **Reviewer XuY2** regarding the baseline selection in the LOGO experiments, specifically the lack of comparison with mainstream context window scaling / long-context alignment works. We agree that these are great suggestions and a crucial addition to LOGO.

Nevertheless, we believe there seems to be a misunderstanding here. We want to provide a quick overview to clarify the core issue our work addresses for all reviewers: *how to perform long-context alignment on a model that already possesses a long context window*.
 In essence, we aim to leverage the inherent information retrieval capability of the long context model[1] to enhance its generation performance. This means we start from utilizing the existing capabilities of LCMs (long context windows) to activate missing capabilities (misalignment).

We intentionally did not compare our method with previous works like LongAlign[2], as these primarily focus on long-context alignment while simultaneously expanding the context window size of models. Their emphasis is on data construction, whereas our starting point is different: we assume the model already has a long context window and aim to improve its alignment and generation capabilities from there. The experimental setups are also distinct, focusing on different aspects of the long-context model's performance.

---

### **2. Criteria for Selecting Baselines and More Evaluation Results**
We note the **Reviewer Bjw9, XuY2, D3My** concern about baseline comparison. In fact, we have already compared LOGO with traditional SFT methods in our paper's tables and figures (Table 1, Figure 4, Figure 5, Figure 7). For clearer comparison, we used the settings recommended by **Reviewer D3My** to reorganize the results as well as demonstrate LOGO's superiority.

We analyze comparisons across two types of tasks: real-world and synthetic, using LongBench and reporting average scores. We select Llama3-8B-Instruct-80K as the backbone model. It is worth noting that the data in the table below originates from Table 1 in the manuscript. Additionally, we also report the results for PoSE[3] and SimPO[4] here:

| Model | Stage | S-DocQA | M-DocQA | Summ | Few-shot | Synthetic | Avg |
|:---|:---:|:---:|:---:|:---:|:---:|:---:|:---:|
| Llama-3-8B-Instruct-8K | - | 39.3 | 36.2 | 24.8 | 63.5 | 39.9 | 40.7 |
| Llama-3-8B-Instruct-80K [1] | CWS | 43.0 | 39.8 | 22.2 | 64.3 | 46.3 | 42.3 |
| &nbsp;&nbsp;&nbsp;&nbsp; + LongAlpaca (SFT) [2] | LA | 39.3 | 36.2 | 26.8 | 63.5 | 48.0 | 42.8 |
| &nbsp;&nbsp;&nbsp;&nbsp; + PoSE (SFT) [3] | LA | 34.9 | 31.4 | 18.7 | 59.3 | 44.2 | 37.7 |
| &nbsp;&nbsp;&nbsp;&nbsp; + SimPO (RL) [4] | LA | 43.2 | 40.7 | 23.5 | 66.7 | 48.4 | 44.5 |
| &nbsp;&nbsp;&nbsp;&nbsp; + LOGO (Ours) | LA | **44.0** | **41.2** | **28.1** | **68.6** | **53.0** | **47.0** |

**Key Findings**
- **Challenges with Continuing SFT Training**: Continuing to train a well-performing Long-context Model with Long-instruction data (SFT) yields minimal benefits and requires a substantial amount of high-quality long-context data to achieve better results. While this observation is not directly related to the claims of our paper, it highlights a broader challenge that necessitates collective efforts from the entire research community.

- **Strategy of Positional Indice Synthesis is important**: Previous work that introduced position synthesis strategy during Long-context alignment (PoSE) causes performance loss due to the gap between position synthesis training and the position indices used in actual model inference - specifically whether all position encodings are visible. Using only skip-wise position encoding synthesis leads to significant performance drops (as indicated in the Table above). In our paper, we introduce a novel Positional Indice Synthesis strategy, which can be referred to Appendix D.

- **Comparison between SimPO and LOGO**, LOGO achieves better results compared with SimPO primarily because it not only increases the space for rejecting dis-preference samples but also adds a CE Loss term to stabilize modeling capability. In long-text alignment tasks, selecting dis-preference samples is challenging (we specifically discuss this point with Reviewer XuY2), and using suboptimal dis-preference samples for SimPO training affects training results.

---

> ### Author Response · Authors · 2024-11-16
> **General Response (Part II)**
>
> ### **3. Manuscript Revision (Important)**
> We have submitted a **revised** version of the manuscript to address reviewers' concerns. All changes made in the manuscript are highlighted in blue font,  and **the rest of the manuscript remains unchanged.**
>
> The modification includes:
>  1) Reorganize the results and add one baseline in Table 1 for a clearer demonstration of our experimental settings;
>  2) Appendix F: Error Bound Analysis;
>  3) Appendix E: Convergence Property from a Gradient Analysis Perspective;
>  4) Appendix H: Error Analysis
>
> **We kindly ask all reviewers to give special consideration to these revisions.**
>
> ---
>
> ### **Reference**
> [1] Wu, Wenhao, et al. "Retrieval head mechanistically explains long-context factuality." arXiv preprint arXiv:2404.15574 (2024).
>
> [2] Bai, Yushi, et al. "Longalign: A recipe for long context alignment of large language models." arXiv preprint arXiv:2401.18058 (2024).
>
> [3] Zhu, Dawei, et al. "PoSE: Efficient Context Window Extension of LLMs via Positional Skip-wise Training." The Twelfth International Conference on Learning Representations.
>
> [4] Meng, Yu, Mengzhou Xia, and Danqi Chen. "Simpo: Simple preference optimization with a reference-free reward." arXiv preprint arXiv:2405.14734 (2024).

---

### Author Response · Authors · 2024-11-21
**Kindly Remind for Follow-up on Submitted Response**

Dear Reviewers,

This is a kind reminder regarding the response we submitted. It has been some time since our first response, and we wonder if any aspects require further clarification or discussions based on our response.

We sincerely appreciate your valuable feedback and remain available to address any further concerns.

Best regards,
Authors

---

### Meta-Review · Area_Chair_kPMm · 2024-12-18

**Metareview:**

(a) The paper introduces LOGO, an efficient preference optimization method for long-context alignment using a reference-free approach and positional index synthesis. The model achieves competitive results with GPT-4 while requiring limited resources.

(b) Strengths include computational efficiency, novel preference optimization, and strong experimental results on long-context tasks.

(c) Weaknesses include incremental contributions, insufficient theoretical justification, and limited evaluation for other baselines or tasks. Some reviewers flagged the need for a stronger analysis of misalignment and scalability.

(d) While promising, the incremental nature and unresolved concerns lead to rejection. The paper lacks rigorous baselines and theoretical depth.

**Additional Comments On Reviewer Discussion:**

During the rebuttal, reviewers highlighted concerns about the lack of theoretical justification, baseline comparisons, and limited scalability analysis. The authors addressed these by adding error analysis, convergence properties, and extended baseline comparisons. Despite improvements, unresolved concerns regarding novelty and broader generalizability influenced the final decision to reject the paper.

---

### Decision · Program_Chairs · 2025-01-22

Reject